# Accurate predictions of protein mutational effects accelerated with a hybrid-topology free energy protocol

Lucien Koenekoop[1], Nadine van de Brug [1,5], Willem Jespers [2,3], Johan Åqvist[1] & Hugo Gutiérrez-de-Terán [1,3,4] ✉

Quantifying the effects of point mutations is of utmost interest for pharmaceutical and biotechnological applications. Reliable computational methods range from statistical and AI-based to physics-based approaches, with the optimal balance between accurate and fast predictions remaining a challenge. Free energy perturbation (FEP) simulations, a powerful physics-based approach available for decades, constitutes nowadays a method of common application in protein mutational studies. We present QresFEP-2, a novel hybrid-topology FEP protocol benchmarked on a comprehensive protein stability dataset of 10 protein systems, encompassing almost 600 mutations. QresFEP-2 combines excellent accuracy with the highest computational efficiency among available FEP protocols, and its robustness is further validated through comprehensive domain-wide mutagenesis, assessing the thermodynamic stability of over 400 mutations generated by a systematic mutation scan of the 56-residue B1 domain of streptococcal protein G (Gβ1). We also demonstrate the applicability domain of QresFEP-2 on evaluating site-directed mutagenesis effects on protein-ligand binding, tested on a GPCR, as well as on protein-protein interactions examined on the barnase/barstar complex. QresFEP-2 emerges as an open-source, physics-based alternative for advancing protein engineering, drug design, and elucidating the impact of mutations on human health.

Understanding the relationships between protein sequence, structure and function is essential for engineering novel biomolecules for industrial and medical applications, as well as fine-tuning the pharmacological regulation of protein targets, in the paradigm of personalized medicine. Single-point mutations can cause alterations in protein structure or function, and potentially manifest as phenotypic changes or even contribute to pathogenesis. Many genetic disorders are caused by missense mutations where a single amino acid substitution leads to abnormal protein function and misfolding, like sickle-cell disease and Rett syndrome[1,2], while complex neurodegenerative conditions like Alzheimer's and Parkinson's disease exemplify the impact of protein mutations on human health[3,4]. Accurate prediction of the effects of point mutations on protein stability not only provides valuable insights into the relationship between evolutionary constraints on protein sequences and their structural properties, but also aids in elucidating the connection between molecular structure and human

disease[5,6]. Furthermore, reliable protein stability predictions offer significant potential for engineering novel protein biocatalysts, for example, by enhancing thermostability or maintaining stability while optimizing other properties (e.g., affinity, solubility, aggregation, or viscosity)[7–10].

Recent advances in structural determination techniques with atomic resolution, such as cryo-EM, have been importantly complemented by the accurate computational predictions enabled by deep-learning approaches like AlphaFold[11,12]. The current comprehensive perspective of protein structures highlights the critical importance of understanding protein stability, which is essential for their biological function and for improving the efficacy of therapeutics targeting them. Within this context, quantitative computational modeling of the effects of point mutations on protein stability or ligand binding is gaining increased importance in protein and ligand design. While numerous computational tools have been developed to predict the effect of mutations on protein stability, significant challenges

[1]Department of Cell & Molecular Biology, Uppsala University, Biomedical Center, Uppsala, Sweden. [2]Medicinal Chemistry, Photopharmacology and Imaging, Groningen Research Institute of Pharmacy, Groningen, AV, The Netherlands. [3]MODSIM Pharma AI B.V., Taxushaag 9, Voorhout, AB, The Netherlands. [4]Nano-materials and Nanotechnology Research Center (CINN), CSIC-University of Oviedo-Principado de Asturias, and Health Research Institute of Asturias (ISPA), Av. del Hospital Universitario, Oviedo, Asturias, Spain. [5]Present address: Department of Computer Science, Faculty of Science and Network Institute, Vrije Universiteit of Amsterdam, Amsterdam, The Netherlands. ✉e-mail: h.g.teran@cinn.es; hugo.gutierrez@icm.uu.se

persist[10,13,14]. Following the promising balance between accuracy and computational efficiency of previous traditional statistical methods, such as FoldX[15], recent years have seen a surge in machine learning approaches[16]. These methods represent a clear advance, yet they face limitations in generalizability, often exhibiting reduced accuracy when applied to novel protein systems beyond their training data[17]. In addition, the effect of protein dynamics or the influence of solvent interactions, both of which can significantly affect protein stability predictions, are usually neglected[18]. This emphasizes the need for more accurate and robust computational approaches for predicting the effects of mutations on protein stability. Here, one can encounter popular structure-based methods such as energy minimization and molecular mechanics Poisson-Boltzmann surface area (MM-PBSA) calculations, which often lack sufficient accuracy to capture the subtle energetic changes associated with single-point mutations[19].

Ideally, predictions of protein stability changes induced by point mutations should reflect the underlying physics of protein folding by accurately modeling the potential energy surface. A rigorous determination of the associated free energies through statistical thermodynamics is achievable using the free energy perturbation (FEP) method, pioneered in this field by Kollman several decades ago[20]. A classical FEP implementation relies on a single molecular topology to describe the two species being compared, where only the changing atoms and their associated parameters are transformed along the perturbation pathway. This approach formed the basis of our original FEP protocol for alanine scanning, which involved the single-topology, stepwise annihilation of amino acid side chains to a common alanine methyl group[21,22]. The implementation of this protocol to non-alanine mutants required such annihilation in parallel simulations of both wild-type (*wt*) and mutant (*mut*) versions of the protein, defining two thermodynamic cycles linked through the common alanine intermediate[23]. Initially developed to characterize and design mutagenesis studies of ligand binding to G protein-coupled receptors (GPCRs), a generalized and fully automated protocol was implemented in the QresFEP software benchmarked against thermal stability data of T4 lysozyme (T4L) or site-directed mutagenesis data on GPCRs[24]. Since such single-topology annihilation avoids changes in atom types or bonded parameters, the protocol emerged as robust and suitable for broad applicability. However, drawbacks of that approach include the potential artifacts caused by the explicit consideration of unnatural alanine intermediate, and the large number of steps required for a converged side-chain annihilation, which is doubled for non-alanine mutations. Other FEP or analogous thermodynamic integration (TI) protocols have been developed and optimized for assessing the effect of side-chain mutations, and usually benchmarked in their ability to predict protein stability changes upon mutation. In particular, the GROMACS-based PMX protocol has been originally published with performance data on the ribonuclease Barnase dataset[14], while the commercial FEP+ from Schrödinger has been validated using a broad dataset encompassing 10 protein targets[10]. Both methodologies employ a dual-topology model for the alchemical transformation of side chains and utilize a full-protein embedding under periodic boundary conditions (PBC) for the associated MD sampling, differing in their specific sampling strategies and other simulation details.

We herein present QresFEP-2, a computationally efficient, dual-topology version of our previous protocol designed to overcome the limitations of the single-topology approach discussed above. The new protocol is versatile and suitable for a range of applications, including the calculation of hydration free energies, protein thermostability, protein-protein interactions, and shifts in ligand-binding affinity induced by protein mutations. Similar to its predecessor, QresFEP-2 is integrated with the molecular dynamics (MD) software Q[25], making it compatible with a number of force fields and taking the advantage of the characteristic spherical boundary conditions used therein, which in combination with the new dual-topology approach maximizes computational efficiency without compromising predictive performance.

After initial validation on hydration free energies of protein side chains, the protocol was calibrated on the T4L dataset and a full benchmark followed on the datasets used for the validation of other FEP protocols, allowing a comparative analysis that shows QresFEP-2 a very accurate and most computational efficient protocol. It followed an original test case on a comprehensive domain-wide mutagenesis dataset, systematically covering a wide range of mutations along the entire sequence of a small 56-residue protein[26]. Finally, we demonstrate the applicability of QresFEP-2 in the drug-discovery domain through a dataset of 26 site-directed mutagenesis experiments on the $A_{2A}$ adenosine receptor ($A_{2A}AR$), a GPCR previously studied with our original side-chain annihilation protocol[22,23], as well as on 11 mutants of the barnase/barstar protein-protein interaction complex. The wide applicability domain, high accuracy and computational efficiency makes Q-resFEP-2 an attractive physics-based approach for the high-throughput virtual screen of protein mutations.

## Results and Discussion
### QresFEP-2: a Hybrid Topology Approach for Automated Residue FEP

QresFEP-2 is an automated, physics-based approach designed to accurately estimate relative free energy changes resulting from protein single-point mutations. The protocol implemented in QresFEP-2 connects separate representations of the *wt* and *mut* side chains through molecular dynamics (MD) sampling along the FEP pathway, defining an implementation of dual topology that we denominate hybrid topology. It thus represents a step forward in computational efficiency as compared to its precursor, QresFEP-1, based on single-topology, gradual annihilation of both *wt* and *mut* side chains to a common alanine methyl group, performed in separate FEP simulations reconvened by joining the resulting thermodynamic cycles[24]. However, according to the definition of single and dual-topology proposed by Ries et al.[27], this distinction requires further nuance. As illustrated in Fig. 1A, a true dual-topology approach would entail separate coordinate sets for the backbone atoms as well, resulting in redundant backbone transformation that would potentially affect the main-chain conformation. Instead, QresFEP-2 utilizes a hybrid topology approach, combining a single-topology representation of the conserved backbone atoms, with a separate (or dual) topology for the variable side-chain atoms. Hybrid topologies for most residue mutations are not univocal; instead they can be defined in various ways as illustrated for the leucine-to-isoleucine mutation in Fig. 1B. On one side of the spectrum, one could design "single-like," mutation-specific pairwise protocols that maintain a single-topology representation on equivalent atoms to maximize phase-space overlap between the side chains, analogous to the maximum common substructure used in most ligand FEP protocols[28]. However, a practical issue arises when atoms that can occupy the same topological (and often spatial) position in their respective side chains belong to different atom types, as illustrated in Fig. 1C. This scenario leads to variations not only in atom type but also in the associated bonded parameters between the end-states, a usual source of problems in terms of convergence as well as for automation purposes. QresFEP-2 instead adopts a "dual-like" hybrid topology approach that combines a single-topology representation for the backbone atoms with separate topologies for all atoms within the side chains (Fig. 1B). Consistent with the philosophy of our previous single-topology QresFEP-1[24] and our dual-topology protocol for ligand FEP simulations (QligFEP)[29], QresFEP-2 avoids transformation of atom types or any bonded parameters (i.e., the set of atoms with associated parameters representing *mut* gradually replaces the *wt* set of atoms and parameters), enabling a rigorous and automatable FEP protocol that exhibits the most "dual-like" character possible.

However, some form of restraint must be imposed between topologically equivalent atoms during the FEP transformation. This is necessary to ensure sufficient phase-space overlap while allowing adequate conformational freedom and preventing alternative erroneous overlap with non-equivalent neighboring atoms, a phenomenon known as "flapping"[30]. QresFEP-2 dynamically addresses this problem, with a double criterion that combines topological equivalence with spatial overlap. Following an initial enumeration of the analogous heavy atoms between the two side chains, these atoms are progressively designated as "restrained to each other" if they are placed within 0.5 Å of each other in their initial conformation. If subsequent analogous heavy atoms (further along the topology of the side chains from the common Cα) have either different atom types or are separated by more than 0.5 Å, whichever occurs first, no further pairwise distance restraints are applied. This

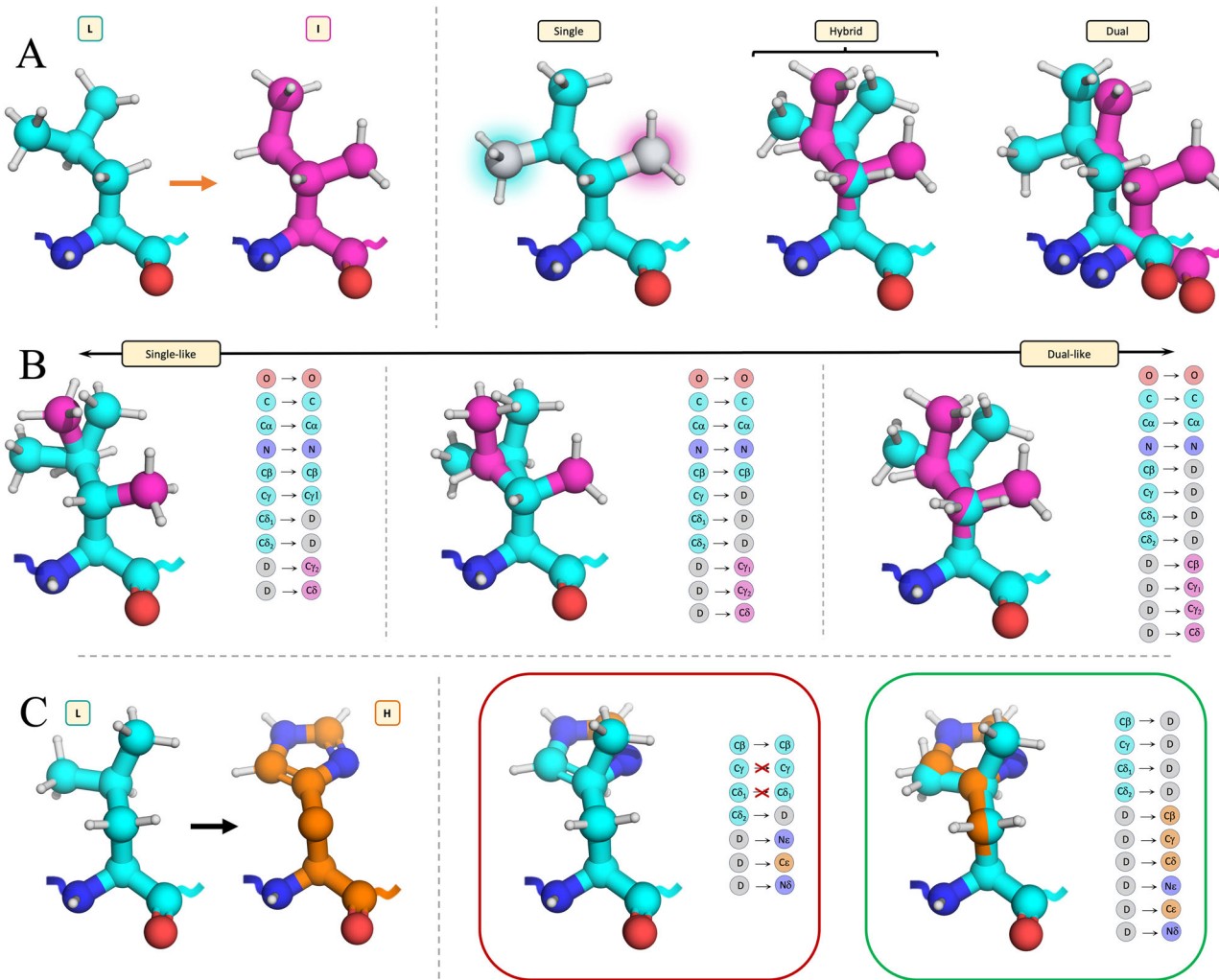

**Fig. 1 | Dual, single and hybrid topologies. A** Different FEP schemes illustrated for the Leu (cyan) → Ile (magenta) transformation: single-topology FEP is based on a unique set of coordinates for both residues, with the gray atoms being transformed indicated with cyan and magenta glow. Dual (or separate) topology FEP employs separate coordinates for all atoms, including the backbone. In between, one example of the Q-resFEP-2 hybrid topology representation based on a common backbone representation and separate side-chain coordinates. **B** The side-chain overlap defines the hybrid topology scheme applied. From left to right, the same Leu → Ile transformation going from a single-like topology (maximizing the overlap of equivalent atoms until Cγ) to a dual-like topology (with overlap assumed only for backbone until Cα). **C** The hybrid topology scheme also depends on the nature of the side chains being perturbed. A Leu → His (orange) transformation is only possible minimizing the side-chain overlap until Cα (right) since a more conservative overlap retaining the equivalence of Cγ and Cδ atoms leads to inconsistencies in atom types, hybridization states, and bonded parameters.

strategy not only simplifies the automated preparation of FEP input files but also yields reliable and accurate results, as demonstrated in the Tyr → Phe test case transformation presented in Fig. 2 and Supplementary Movie 1. One can observe that the minimum restraining scheme required for accurate results affects relative position of the two side chains until the initial common atoms of the ring (Cγ). While the results are not very sensitive to additional further restraints, we retained the automated dynamic definition of the restraining scheme for the remainder of this work.

## The QresFEP-2 thermodynamic cycle for protein stability

Experimental changes in protein stability due to a single-point mutation are usually reported as a free energy change ($\Delta\Delta G$, kcal·mol$^{-1}$). One can indeed model such changes as the corresponding difference in protein folding energy between the *wt* and *mut* versions of the protein with the unfolded state represented by a reference tripeptide, thus defining a thermodynamic cycle that can be solved as depicted in Fig. 3.

In QresFEP-2, two analogous *wt* → *mut* FEP simulations are setup and run in parallel, each accounting for different environments. In the folded state, typically extracted from a PDB of the protein, a solvated sphere is centered on

the mutable residue, while the unfolded state is modeled with the mutable residue as the central position of a tripeptide[31,32] using a similar spherical solvation model[33]. The influence of the flanking residues in the reference tripeptide model (i.e., natural sequence, alanine, glycine, or a single capped residue) is investigated later in this work. Both simulations must be performed under identical conditions to ensure consistency throughout the thermodynamic cycle, which means adopting identical hybrid topology schemes and associated restraints as defined above. It follows that the results of an alanine-based tripeptide simulation cannot be simulated once and stored for later use, as has been done in previous protocols by us and others[10,14,24]. Instead, QresFEP-2 dynamically defines and applies on the reference state the necessary set of distance-restraints for a given side-chain comparison, on the basis of the relative positions of equivalent atoms in the modeled *wt* and *mut* protein structures. Finally, the full FEP (*wt* → *mut*) transformation is divided in two consecutive stages: i) a gradual "turn-off" of the atomic charges corresponding to *wt* side-chain atoms, coupled with the introduction of a soft-core potential on the van der Waals terms of both *wt* and *mut* side-chain atoms; and ii) a gradual "turn-on" of the atomic charges corresponding to *mut* side-chain atoms, coupled with the removal of the soft-core potentials defined above.

**Fig. 2 | Hybrid topology FEP transformation of mutant Y24F from Ribonuclease Barnase.** The *wt* residue (Tyr) is shown as cyan, and the *mut* (Phe) in magenta, in both cases as ball and sticks, with explicit representation of neighboring residues and water molecules within 4 Å. The table and the schematic 2D representation of the Y24F mutation show the results using different restraining schemes: from no restraints (-), showing the most dual-like character, to the maximum pairwise restraining scheme (Cζ), including all topologically equivalent and special overlapped atoms, and a single-like, hybrid topology FEP transformation included for comparison. Error estimates between the experimental and predicted ΔΔ*G* are shown to indicate the accuracy, together with standard error of the mean (SEM) values to illustrate the precision of the calculations, both in kcal·mol⁻¹.

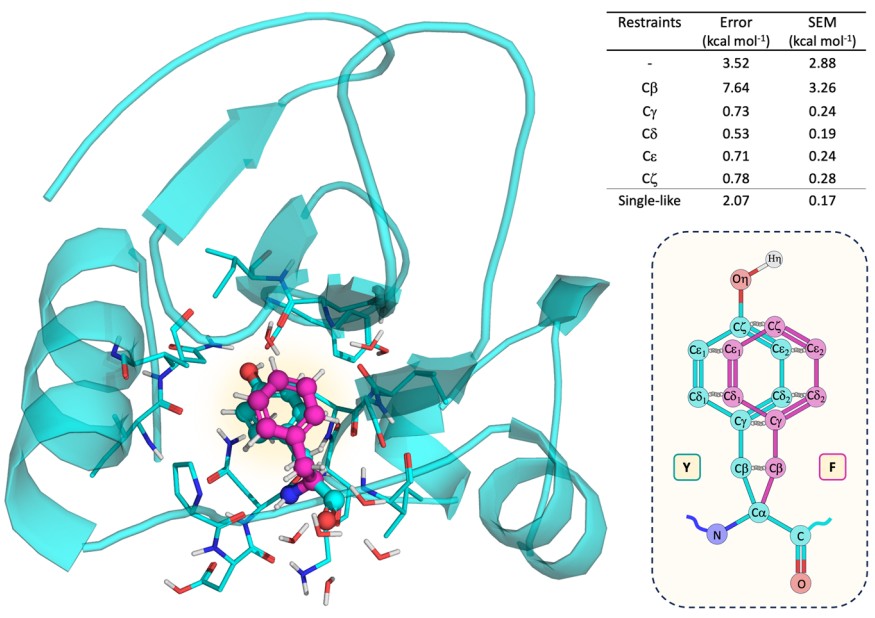

| Restraints | Error (kcal mol⁻¹) | SEM (kcal mol⁻¹) |
|---|---|---|
| - | 3.52 | 2.88 |
| Cβ | 7.64 | 3.26 |
| Cγ | 0.73 | 0.24 |
| Cδ | 0.53 | 0.19 |
| Cε | 0.71 | 0.24 |
| Cζ | 0.78 | 0.28 |
| Single-like | 2.07 | 0.17 |

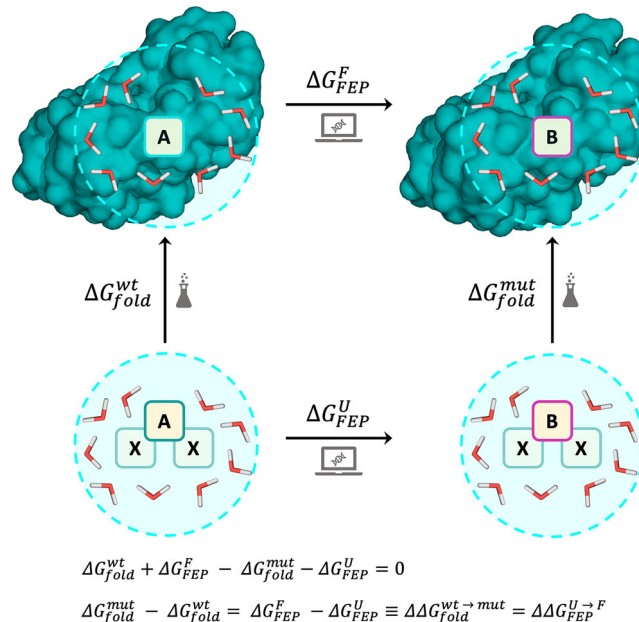

$$\Delta G_{fold}^{wt} + \Delta G_{FEP}^{F} - \Delta G_{fold}^{mut} - \Delta G_{FEP}^{U} = 0$$

$$\Delta G_{fold}^{mut} - \Delta G_{fold}^{wt} = \Delta G_{FEP}^{F} - \Delta G_{FEP}^{U} \equiv \Delta\Delta G_{fold}^{wt \rightarrow mut} = \Delta\Delta G_{FEP}^{U \rightarrow F}$$

**Fig. 3 | Thermodynamic cycle used in QresFEP-2.** Both horizontal legs represent the free energy perturbations of wild-type residue (**A**) to mutant residue (**B**), which include transformations performed on the solvated protein folded state (F, top), and analogous transformation performed on a tripeptide in solution representing the unfolded state (U, bottom). Vertical legs account for experimentally determined folding energies of the *wt* (residue A, left) and the *mut* (residue B, right) protein versions.

Each side chain (*wt* or *mut*) will have disappearing atoms in the respective end-state, which gradually transition along the transformation to dummy atoms that only interact through bonded terms. The result of this approach is equivalent contributions to free energy differences that effectively cancel out in the thermodynamic cycle[30]. Throughout this process, the pairwise non-bonded interactions between the side-chain atoms of the *wt* and *mut* residues are excluded from the calculations.

## Hydration free energies
The dataset of experimental hydration free energies of amino acid side-chain mimics, reported by Wolfenden et al.[34], has become a common

benchmark in the field of force field development and free energy calculations. Different FEP/TI methods have evaluated their predictive power by determining the relative hydration energy of each side-chain mimic with respect to methane, for which the water solvation energy was experimentally measured as a mimic of the side chain of Alanine[34], using a thermodynamic cycle that compares the perturbations of the molecules of interest in vacuum and water, respectively. Table 1 presents the experimental and QresFEP-2 calculated results for the hydration free energies, relative to methane, for all side-chain analogs excluding proline, glycine and all titratable residues. Our calculations demonstrate excellent agreement with experimental data, with a quantitative accuracy expressed in terms of the mean absolute error (MAE) of 0.33 kcal·mol⁻¹ and a correlation coefficient ($R^2$) of 0.99. The simulations also showed excellent statistical convergence, reflected in a standard error of the mean (SEM) of the average ΔΔ*G* values (as calculated from 10 independent replica simulations) not exceeding 0.10 kcal·mol⁻¹ in any case. As shown in Supplementary Fig. 1, these results outperform those obtained with our single-topology approach QresFEP-1 (MAE = 0.85 kcal·mol⁻¹; $R^2$ = 0.94) and with the related dual-topology protocol for ligand perturbations, QligFEP (MAE = 0.95 kcal·mol⁻¹; $R^2$ = 0.91)[29], in all cases using the OPLSAA/M force field[35]. No significant differences were observed between the two alternative methods implemented for calculating relative free energies, i.e., Zwanzig exponential formula or Bennet Acceptance Ratio (BAR), and we will report BAR analysis throughout this study consistent with previous findings for amino acid[29] and pair-base mutations[36]. The performance of QresFEP-2 was compared to other published methods beyond QresFEP-1 (Supplementary Table 1). QresFEP-2 ranks as the second most accurate protocol for neutral residues, outperformed only by results obtained with the GROMOS 53a6 force field[37], which was specifically tailored for calculating hydration free enthalpies of amino acids.

## Protein stability: alanine scan
Once the basic hybrid-topology QresFEP-2 protocol was verified with the side-chain hydration free energies benchmark, we proceeded with estimation of protein stability changes using available experimental datasets. Initially investigation of the stability effects of 43 selected alanine mutations of T4 lysozyme (T4L) from the ProTherm database[38], was carried out with a dual purpose: (i) to optimize the parameters of the FEP protocol and (ii) to directly compare the accuracy and efficiency of the hybrid-topology QresFEP-2 with the previous single-topology QresFEP-1 protocol[29], as well as with the widely used Schrödinger's FEP + [10].

**Table 1 | Experimental and calculated hydration free energies of amino acid side-chain mimics (X) relative to methane (Me)**

| Amino acid | Side-chain mimic | $\Delta\Delta G_{exp}^{solv}$ (kcal·mol$^{-1}$) | X → Me (BAR) (kcal·mol$^{-1}$) | X → Me (Zwanzig) (kcal·mol$^{-1}$) |
|---|---|---|---|---|
| Asparagine | Acetamide | 11.62 | 11.18 ± 0.06 | 11.18 ± 0.06 |
| Cysteine | Methanethiol | 3.18 | 2.77 ± 0.02 | 2.77 ± 0.02 |
| Glutamine | Propionamide | 11.32 | 11.46 ± 0.07 | 11.46 ± 0.07 |
| Isoleucine | 1-Butane | −0.21 | 0.13 ± 0.05 | 0.13 ± 0.05 |
| Leucine | Isobutane | −0.34 | −0.04 ± 0.02 | −0.03 ± 0.02 |
| Methionine | Methylsulfanylethane | 3.42 | 2.67 ± 0.06 | 2.67 ± 0.06 |
| Phenylalanine | Toluene | 2.70 | 3.47 ± 0.04 | 3.47 ± 0.04 |
| Serine | Methanol | 7.00 | 6.99 ± 0.05 | 6.98 ± 0.05 |
| Threonine | Ethanol | 6.82 | 6.91 ± 0.10 | 6.91 ± 0.10 |
| Tryptophane | 3-Methyl-1H-indole | 7.82 | 7.72 ± 0.08 | 7.72 ± 0.08 |
| Tyrosine | p-Cresol | 8.05 | 8.42 ± 0.07 | 8.43 ± 0.06 |
| Valine | Propane | −0.05 | 0.08 ± 0.02 | 0.08 ± 0.02 |
| | | | $R^2 = 0.99\ ^{1.00}_{0.97}$ | $R^2 = 0.99\ ^{1.00}_{0.97}$ |
| | | | $MAE = 0.33\ ^{0.48}_{0.20}$ | $MAE = 0.32\ ^{0.46}_{0.19}$ |
| | | | $\rho = 0.96\ ^{1.00}_{0.77}$ | $\rho = 0.96\ ^{1.00}_{0.78}$ |
| | | | $\tau = 0.85\ ^{1.00}_{0.61}$ | $\tau = 0.85\ ^{1.00}_{0.60}$ |

Particular FEP values are presented as average ± SEM obtained from 10 replicate simulations (see text). The statistical figures of merit along this work are presented as average values with 95% confidence intervals (CI)[60].

$R^2$ coefficient of determination, $MAE$ mean absolute error, $\rho$ Spearman's rank correlation coefficient (quantifies how well the relationship between two variables can be described using a monotonic function), $\tau$ Kendall rank correlation coefficient (measures the degree of similarity between the orderings of two sets of ranks).

**Table 2 | Statistical analysis of the different FEP protocols applied to the T4L dataset**

| Method | T4L mutations | $n$ | MAE (kcal·mol$^{-1}$) | Accuracy (%) | MCC | $R^2$ | $\rho$ | $\tau$ |
|---|---|---|---|---|---|---|---|---|
| QresFEP-2 (Protocol H) | Ala-scan | 43 | $1.26\ ^{1.51}_{1.03}$ | 93.0 | $0.76\ ^{1.00}_{0.48}$ | $0.72\ ^{0.85}_{0.54}$ | $0.86\ ^{0.93}_{0.75}$ | $0.69\ ^{0.80}_{0.57}$ |
| QresFEP-2 (Protocol J) | Ala-scan | 43 | $1.09\ ^{1.32}_{0.88}$ | 88.4 | $0.57\ ^{0.87}_{0.18}$ | $0.73\ ^{0.85}_{0.58}$ | $0.88\ ^{0.93}_{0.77}$ | $0.71\ ^{0.81}_{0.60}$ |
| QresFEP-1[24] | Ala-scan | 43 | $1.44\ ^{1.71}_{1.18}$ | 83.7 | $0.49\ ^{0.78}_{0.10}$ | $0.74\ ^{0.84}_{0.60}$ | $0.87\ ^{0.93}_{0.76}$ | $0.71\ ^{0.80}_{0.59}$ |
| FEP+[10] | Ala-scan | 43 | $1.18\ ^{1.44}_{0.92}$ | 88.4 | $0.48\ ^{0.85}_{-0.06}$ | $0.69\ ^{0.82}_{0.50}$ | $0.81\ ^{0.89}_{0.66}$ | $0.62\ ^{0.74}_{0.48}$ |
| QresFEP-2 (Protocol H) | All | 66 | $1.41\ ^{1.68}_{1.16}$ | 84.9 | $0.55\ ^{0.77}_{0.28}$ | $0.59\ ^{0.78}_{0.36}$ | $0.73\ ^{0.87}_{0.53}$ | $0.58\ ^{0.71}_{0.43}$ |
| QresFEP-2 (Protocol J) | All | 66 | $1.28\ ^{1.54}_{1.05}$ | 83.3 | $0.40\ ^{0.68}_{0.08}$ | $0.60\ ^{0.78}_{0.38}$ | $0.74\ ^{0.86}_{0.54}$ | $0.57\ ^{0.70}_{0.44}$ |
| FEP+[10] | All | 66 | $1.21\ ^{1.46}_{0.97}$ | 84.9 | $0.36\ ^{0.67}_{0.03}$ | $0.70\ ^{0.81}_{0.55}$ | $0.81\ ^{0.88}_{0.69}$ | $0.61\ ^{0.70}_{0.50}$ |

Accuracy expressed as % of correct predictions.

$n$ number of mutations, $R^2$ coefficient of determination, $MAE$ mean absolute error, $\rho$ Spearman's rank correlation coefficient, $\tau$ Kendall rank correlation coefficient, $MCC$ Matthews correlation coefficient.

To optimize the protocol, we examined the influence of different FEP simulation parameters on a subset of six T4L residues, representing diverse sizes, protein location and chemical properties. A total of 10 protocols (A-J) were tested, each of them with different combinations of the following sampling parameters: number of λ-windows (20, 50, 100), sampling time (10 or 20 ps/λ), and the sampling scheme (linear vs sigmoidal, having evenly or unevenly spaced λ-windows, in the latter case with higher density towards the end-states)[39]. The results are shown in Supplementary Fig. 2, where it can be observed that protocol H provides an optimal trade-off between MD sampling, accuracy (MAE) and robustness (SEM), which were the three criteria under evaluation. In this protocol, each of the two consecutive FEP stages consists of 50 λ-steps with a sampling time of 20 ps per step,

distributed along a sigmoidal sampling path. The overall sampling time, considering the 10 independent replicate simulations, is thus 20 ns for each leg of the thermodynamic cycle (i.e., folded or unfolded state simulations) per mutation (Fig. 3, Supplementary Fig. 2).

Using this protocol, we calculated the effect of the 43 alanine mutations on the T4L dataset. The results are summarized on Table 2 and detailed in Supplementary Table 2, while Supplementary Fig. 3 shows the high correlation between these results and those previously obtained with the single-topology successive annihilation protocol QresFEP-1 ($R^2 = 0.83$, MAE = 0.84). A slight, statistically non-significant improvement in the predictions could be observed with the hybrid-topology QresFEP-2 (Table 2, Protocol H), while the computation time is reduced by 2 to 4–fold depending on the

**Table 3 | Reference tripeptide analysis for Barnase ribonuclease and T4 lysozyme datasets**

| Protein | PDB ID | Method | Tripeptide[a] | n | MAE (kcal · mol$^{-1}$) | Accuracy (%) | MCC | $R^2$ | $\rho$ | $\tau$ |
|---|---|---|---|---|---|---|---|---|---|---|
| Barnase | 1BNI | QresFEP-2 | cZXZc | 109 | $0.83^{0.99}_{0.70}$ | 88.1 | $0.35^{0.61}_{0.05}$ | $0.66^{0.79}_{0.49}$ | $0.70^{0.81}_{0.57}$ | $0.54^{0.65}_{0.43}$ |
| Barnase | 1BNI | QresFEP-2 | cAXAc | 109 | $0.81^{0.96}_{0.67}$ | 87.2 | $0.33^{0.59}_{0.03}$ | $0.70^{0.81}_{0.54}$ | $0.74^{0.84}_{0.62}$ | $0.59^{0.69}_{0.47}$ |
| Barnase | 1BNI | QresFEP-2 | cGXGc | 109 | $0.86^{1.01}_{0.73}$ | 82.6 | $0.26^{0.49}_{0.00}$ | $0.72^{0.82}_{0.57}$ | $0.77^{0.85}_{0.65}$ | $0.60^{0.68}_{0.49}$ |
| Barnase | 1BNI | QresFEP-2 | cXc | 109 | $1.09^{1.27}_{0.93}$ | 80.7 | $0.17^{0.41}_{-0.08}$ | $0.67^{0.78}_{0.50}$ | $0.69^{0.80}_{0.56}$ | $0.52^{0.62}_{0.41}$ |
| Barnase | 1BNI | PMX[41] | cGXGc | 109 | $0.79^{0.93}_{0.67}$ | 89.0 | $0.42^{0.66}_{0.13}$ | $0.74^{0.85}_{0.55}$ | $0.76^{0.85}_{0.65}$ | $0.59^{0.68}_{0.49}$ |
| Barnase | 1A2P | FEP+[10] | cXc | 55 | $0.83^{1.00}_{0.67}$ | 90.9 | $0.24^{0.70}_{-0.08}$ | $0.59^{0.74}_{0.40}$ | $0.71^{0.84}_{0.50}$ | $0.53^{0.67}_{0.37}$ |
| T4L | 2LZM | QresFEP-2 | cZXZc | 66 | $1.41^{1.68}_{1.15}$ | 84.9 | $0.55^{0.77}_{0.28}$ | $0.59^{0.78}_{0.35}$ | $0.73^{0.87}_{0.53}$ | $0.58^{0.70}_{0.43}$ |
| T4L | 2LZM | QresFEP-2 | cAXAc | 66 | $1.52^{1.80}_{1.26}$ | 81.8 | $0.38^{0.64}_{0.06}$ | $0.54^{0.73}_{0.31}$ | $0.70^{0.83}_{0.51}$ | $0.54^{0.67}_{0.39}$ |
| T4L | 2LZM | QresFEP-2 | cGXGc | 66 | $1.62^{1.93}_{1.32}$ | 81.8 | $0.31^{0.60}_{-0.02}$ | $0.55^{0.76}_{0.31}$ | $0.72^{0.85}_{0.52}$ | $0.55^{0.68}_{0.41}$ |
| T4L | 2LZM | QresFEP-2 | cXc | 66 | $1.65^{2.00}_{1.33}$ | 86.4 | $0.52^{0.77}_{0.21}$ | $0.54^{0.75}_{0.30}$ | $0.72^{0.85}_{0.52}$ | $0.55^{0.68}_{0.42}$ |

Accuracy expressed as % of correct predictions.

n number of mutations, $R^2$ coefficient of determination, MAE mean absolute error, $\rho$ Spearman's rank correlation coefficient, $\tau$ Kendall rank correlation coefficient, MCC Matthews correlation coefficient.
[a]'X' indicates the mutable residue, and 'c' the capped termini.

nature of the side chain being mutated[29]. As an example, mutation of a mid-size amino acid (Ile → Ala) with the single-topology gradual annihilation of QresFEP-1 involves 6 subperturbations x 50 λ-windows x 10,000 (1 fs) steps = 3 M steps, while the same mutation in QresFEP-2 involves 2 stages x 50 λ-windows x 10,000 (2 fs) steps = 1 M steps. Doubling the number of λ steps to match the sampling of the QresFEP-1 protocol (Table 2, QresFEP-2, protocol J) results in non-significant differences in terms of correlation, quantitative or qualitative accuracy, the last expressed both in terms of percentage of correct predictions and as the Matthews correlation coefficient (MCC). When compared to the commercial FEP + [10], the extended protocol J also displays an apparent although non-statistically significant improvement. In this sense, the threshold of experimental accuracy should be considered when comparing the accuracy of different theoretical methods. As a reference, an internal validation study of the experimental data within the ProTherm database revealed an accumulated MAE of 0.81 kcal·mol$^{-1}$, with an internal correlation of $R^2 = 0.71$[40]. Along the rest of this study, QresFEP-2 calculations are performed with protocol H. While we consider this sampling protocol a good starting point of general applicability to study effects of protein mutations, it is important to remind the flexibility of QresFEP-2 in setting up different sampling parameters.

## Protein stability: non-alanine mutations

The hybrid-topology protocol for alanine scanning does not differ substantially from a single-topology annihilation, although the reduction of sampling time needed for convergence was noticeable. However, the main advantages are expected when modeling the effects of non-alanine mutations, where the dual annihilation to a common Ala from both *wt* and *mut* side chains required by QresFEP-1 comes with a cost in both computational efficiency and presumably precision, due to error accumulation[29]. To validate the transferability of the QresFEP-2 optimized protocol to non-alanine mutations, we expanded the T4L benchmark to include the set of 23 non-alanine mutants present in the ProTherm dataset. We initially maintained the sampling protocol H and excluded mutations to/from charged side chains or proline, allowing for comparison with FEP+ results[10]. The overall performance of QresFEP-2 (protocol H, Table 2 and Supplementary Table 2) was again quite satisfactory, falling within the same range of accuracy of FEP+. In line with the observations on the alanine-scan subset,

doubling the sampling time only marginally improved these metrics in a statistically non-significant manner (protocol J, Table 2). Therefore, we maintained protocol H for the remainder of this study as the optimal trade-off between accuracy and computational efficiency.

We then moved on the explore the dataset of bacterial ribonuclease barnase, a well-studied system previously used to benchmark not only the FEP+ protocol[10] but also the GROMACS-based PMX protocol (previously PYMACS)[41]. The full dataset comprises 109 data points, including 31 mutations to alanine, 17 to glycine, and 61 non-alanine/non-glycine mutants (56% of the dataset). The results, presented in Table 3 and in detail in Supplementary Table 3, show satisfactory accuracy and correlation with experiments within the same range as the alternative FEP methods[10,41]. We noted that one difference between these methods is the way that the reference state is modeled. In our previous approach[24], we used the common alanine-based tripeptide (AXA) that allows storing the calculated values in databases for future calculations[32]. Instead, the QresFEP-2 protocol extracts each time the tripeptide from the natural sequence (ZXZ for convention), facilitating automation and allowing the necessary dynamic definition of the restraints of the reference tripeptide as discussed above (Fig. 2). However, the potential influence of the flanking side chains on the tripeptide model remained an open question raised by others[42], which we addressed by evaluating the performance on the T4L and barnase datasets of three additional models: the AXA tripeptide, the GXG tripeptide, which is the reference state model used in PMX[41], and the mutable residue X alone (all capped with the N-methylated and C-acetylated termini in solution), the approach adopted in FEP+[10]. A comparative analysis indicates that the reference state model does not have a statistically significant effect on FEP results, validating the pragmatic choice of the ZXZ tripeptide as the reference state in QresFEP-2 (Table 3).

One other aspect that might influence the FEP results of single-point mutations is the initial conformation of the modeled mutant side chain. In QresFEP-2, the most probable conformation is instantly generated with the mutagenesis tool implemented in PyMOL[43]. Arguably, an alternative approach would be to generate a 3D model of the mutant protein with AlphaFold2 (AF2)[12], a process that requires 2–3 hours to generate 5 ranked models and must be followed by the incorporation of the *mut* side-chain conformation from the AF2 model into the original *wt* protein structure. To

**Table 4 | Benchmark results divided by protein validation set**

| Protein | PDB ID | n | MAE (kcal · mol⁻¹) | Accuracy (%) | MCC | R² | ρ | τ |
|---|---|---|---|---|---|---|---|---|
| T4 lysozyme | 2LZM | 66 | $1.41{}^{1.68}_{1.15}$ | 84.9 | $0.55{}^{0.76}_{0.28}$ | $0.59{}^{0.78}_{0.35}$ | $0.73{}^{0.87}_{0.54}$ | $0.58{}^{0.71}_{0.43}$ |
| Barnase ribonuclease | 1BNI | 109 | $0.83{}^{0.99}_{0.70}$ | 88.1 | $0.35{}^{0.61}_{0.04}$ | $0.66{}^{0.79}_{0.49}$ | $0.70{}^{0.81}_{0.57}$ | $0.54{}^{0.66}_{0.42}$ |
| Staphylococcal nuclease | 1STN | 164 | $1.36{}^{1.60}_{1.14}$ | 88.4 | $0.48{}^{0.67}_{0.26}$ | $0.57{}^{0.69}_{0.46}$ | $0.79{}^{0.85}_{0.72}$ | $0.61{}^{0.67}_{0.53}$ |
| Chymotrypsin inhibitor 2 | 1YPC | 42 | $0.77{}^{0.96}_{0.69}$ | 78.6 | $0.10{}^{0.48}_{-0.16}$ | $0.64{}^{0.77}_{0.47}$ | $0.79{}^{0.89}_{0.62}$ | $0.60{}^{0.72}_{0.45}$ |
| Protein L, B1 domain | 1HZ6 | 44 | $0.99{}^{1.22}_{0.76}$ | 90.9 | $0.31{}^{0.81}_{-0.08}$ | $0.74{}^{0.85}_{0.58}$ | $0.82{}^{0.91}_{0.66}$ | $0.64{}^{0.76}_{0.49}$ |
| c-SRC tyrosine kinase | 1FMK | 40 | $2.18{}^{3.03}_{1.45}$ | 90.0 | $0.62{}^{0.90}_{0.35}$ | $0.26{}^{0.51}_{0.05}$ | $0.52{}^{0.76}_{0.20}$ | $0.40{}^{0.61}_{0.15}$ |
| Human lysozyme | 1REX | 45 | $1.80{}^{2.28}_{1.37}$ | 73.3 | $0.27{}^{0.58}_{-0.07}$ | $0.38{}^{0.65}_{0.10}$ | $0.43{}^{0.69}_{0.13}$ | $0.32{}^{0.52}_{0.09}$ |
| Fibronectin III domain | 1TEN | 29 | $0.97{}^{1.22}_{0.72}$ | 100 | $1.00{}^{1.00}_{1.00}$ | $0.82{}^{0.91}_{0.67}$ | $0.91{}^{0.97}_{0.76}$ | $0.76{}^{0.88}_{0.61}$ |
| Trypsin inhibitor | 1BPI | 17 | $1.28{}^{2.02}_{0.72}$ | 88.2 | *n.a.* | $0.07{}^{0.53}_{0.00}$ | $0.39{}^{0.77}_{-0.13}$ | $0.28{}^{0.63}_{-0.12}$ |
| FK506 binding protein | 1FKB | 27 | $0.93{}^{1.26}_{0.64}$ | 92.6 | $0.46{}^{1.00}_{-0.08}$ | $0.55{}^{0.77}_{0.35}$ | $0.85{}^{0.94}_{0.69}$ | $0.68{}^{0.83}_{0.52}$ |
| **Total (QresFEP-2)** | | **583** | $\mathbf{1.25}{}^{1.36}_{1.14}$ | **87.1** | $\mathbf{0.41}{}^{0.52}_{0.30}$ | $\mathbf{0.49}{}^{0.57}_{0.41}$ | $\mathbf{0.71}{}^{0.76}_{0.66}$ | $\mathbf{0.53}{}^{0.57}_{0.49}$ |
| Total (FEP+)[10] | | 534 | $1.38{}^{1.52}_{1.25}$ | 87.6 | $0.34{}^{0.47}_{0.21}$ | $0.47{}^{0.58}_{0.38}$ | $0.73{}^{0.77}_{0.67}$ | $0.54{}^{0.58}_{0.50}$ |

Accuracy expressed as % of correct predictions.

*n* number of mutations, $R^2$ coefficient of determination, *MAE* mean absolute error, *ρ* Spearman's rank correlation coefficient, *τ* Kendall rank correlation coefficient, *MCC* Matthews correlation coefficient.

evaluate any potential advantage of such a more demanding approach, we performed calculations starting from either PyMOL or AF2-generated conformations on a subset of 14 diverse mutations spanning different regions of barnase. We also evaluated if the latest version AF3 (released during the course of this project) would have impact on the FEP estimations. The results (see Supplementary Fig. 4) clearly show no significant differences between the three sets of calculations, reinforcing our strategy of using the faster and more scalable PyMOL mutagenesis tool as the method for generating mutant residues.

### Expanded benchmark for protein stability

At this point, we expanded the benchmark with 8 additional datasets originally compiled from the FoldX benchmark study[44]. Together with T4L and barnase, the 10 protein systems constituted the full benchmark for the validation of Schrödinger's FEP+ in predicting thermal stability effects of point mutations[10]. These systems correspond to small proteins with less than 170 amino acids (except for c-SRC tyrosine kinase), with at least one high-resolution crystal structure (<2.0 Å), and thermal stability data available for a substantial number of mutations (>33 mutants per protein). After omitting proline and terminal mutations, as well as mutations involving charged residues (i.e., to/from arginine, lysine, aspartic acid, and glutamic acid), we retained 583 data points, categorized by protein system and accounting for at least 17 datapoints per protein. Note that, due to the inclusion of 53 additional mutants for barnase extracted from ref. 42, our dataset is approximately 10% larger than the equivalent subset of "charge-conserved mutations" in the original FEP+ study; other than that, the results reported here are directly comparable to the FEP+ validation study[10].

The overall performance of QresFEP-2 on the entire dataset is detailed in Table 4 and visualized in Fig. 4. The quantitative accuracy and correlation are remarkable, with an overall MAE = 1.25 kcal·mol⁻¹, and $R^2$ = 0.49. One critical aspect of large-scale prediction of protein mutation effects is the ability to correctly classify mutations as stabilizing or destabilizing. A preliminary analysis shows an encouraging trend, with 87% of the mutations predicted with the correct sign. However, this metric can be misleading due

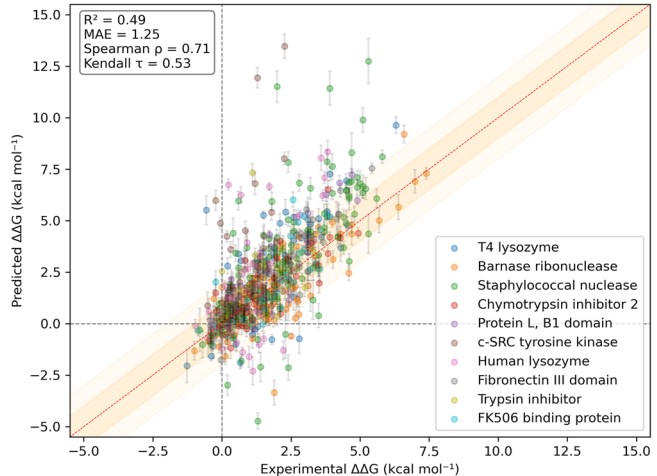

**Fig. 4 | Experimental *vs* calculated free energy shifts of thermal stability.** The plot contains all 583 datapoints included in the complete benchmark (see Table 3), expressed as ΔΔG (kcal·mol⁻¹) and colored by protein system.

to the imbalance in the original dataset, which has a predominance of destabilizing mutations and can artificially inflate the correlation, given an a priori higher probability for a simple model of a single-point mutation to destabilize the original structure. MCC mitigates this bias, providing a more balanced evaluation even with skewed distributions. Since it integrates information from all elements of a confusion matrix, it reflects both correct and incorrect predictions across all classes, and ranges from -1 (perfect anticorrelation) to +1 (perfect correlation) with 0 indicating performance no better than random guessing. The global value obtained of MCC = 0.41 indicates a tendency of the model to predict both stabilizing and destabilizing effects of point mutations across a diverse dataset.

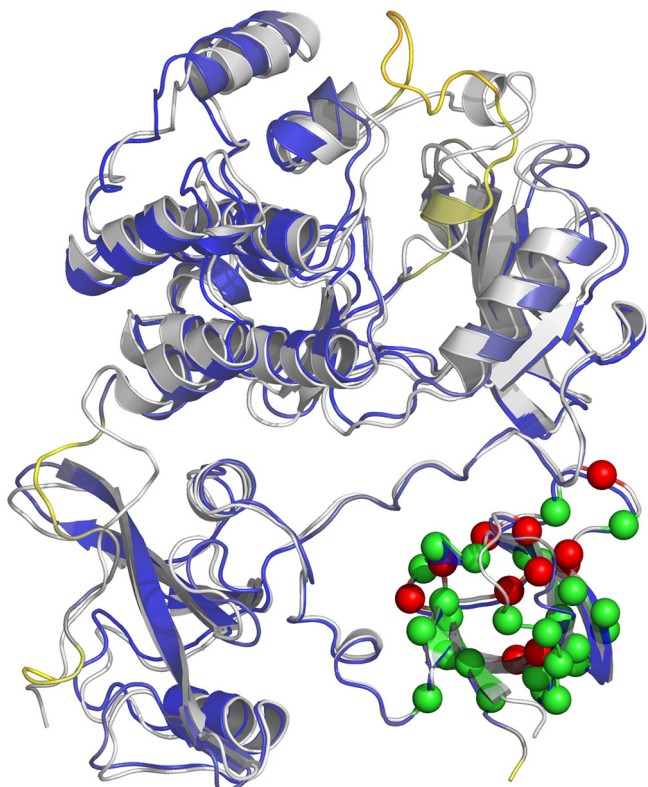

**Fig. 5 | Experimental and modeled structures for c-SRC tyrosine kinase.** The crystal structure is colored gray and the AlphaFold2 model in blue, with yellow indicating low-confident modeled region. Mutations studied are shown as spheres, with green indicating an acceptable error threshold, while red spheres correspond to those mutants identified as outliers within QresFEP-2 predictions (see Fig. 4).

A breakdown of the results by protein system (Supplementary Fig. 5–14, Supplementary Tables 2–11) can provide further insights into the robustness of the method and help identifying problematic datasets and potential limitations of our approach. In terms of quantitative accuracy, we observed satisfactory MAE values ranging from 0.77 to 1.80 kcal·mol$^{-1}$ for all systems, except c-SRC tyrosine kinase (MAE = 2.18 kcal·mol$^{-1}$). Interestingly, the original FEP+ benchmarking also showed a comparably lower accuracy for c-SRC tyrosine kinase, with the two main outliers in both protocols being Y92A and W118A, the biggest outliers in the whole dataset (Fig. 4). Indeed, structural analysis reveals that most outliers, which are overpredicted, are clustered in the same region of the protein (Fig. 5). The relatively narrow range of experimental values likely contributes to the tendency of overprediction observed for this protein dataset (Supplementary Fig. 10), which on the other hand does not compromise the accuracy in the classification of stabilizing/destabilizing mutations (MCC = 0.62). The range of MCC values indicate predictive models in all cases with the only exception of trypsin inhibitor, a dataset consisting on only 17 stabilizing mutations thus precluding MCC assessment. Conversely, the Fibronectin III domain displays perfect correlation (MCC = 1) with zero error margin as arises from the bootstrapping analysis, though this may be influenced by the relatively small size of the dataset and the presence of only one destabilizing mutation (S875A).

## Computational efficiency

Remarkably, QresFEP-2 exhibits an overall improved accuracy as compared to the commercial FEP+ software[10], with MAE differences being statistically significant for the first time along this comparative study (Table 4 and Supplementary Fig. 15, $p < 0.0001$), but at a fraction of the computational cost. As we will see, QresFEP-2 emerges as the most computational efficient FEP protocol available, largely attributed to its utilization of the spherical

boundary SCAAS model for the MD simulations employed in the Q software[25,33], as opposed to the more extensively used periodic boundary conditions (PBC). To analyze the differences in computational performance between methods, one has to remind that the computational CPU/GPU time for MD simulations typically shows a quadratic dependence on the number of atoms in the system. The effect of boundary conditions on this parameter can be easily illustrated with a typical protein system such as c-SRC tyrosine kinase, which exceeds 400 residues: the corresponding PBC box used in most FEP protocols results in ~90.000 atoms, whereas the 50 Å diameter solvated sphere, as defined around the mutable residue in the QresFEP protocols, involves ~7.000 atoms. In terms of computational resources, the sampling required with the QresFEP-2 protocol H (40 ns for the whole thermodynamic cycle, see Supplementary Fig. 2) was completed within 3 hours of wall time using the 160 CPU cores available for this project (HPE Cray EX, AMD EPYC 7742 64 C 2.25 GHz, Slingshot-11). This amounts to 480 CPU hours, with the total associated cost for each mutation of approximately 11 USD. Notably, these numbers are independent of protein size, and are in stark contrasts with the estimations provided by Steinbrecher et al. using FEP+[10]. Therein, a single 5 ns simulation per leg of the thermodynamic cycle for a small system such as Chymotrypsin Inhibitor took 4 hours of wall time using 4 GPUs (Nvidia GeForce GTX780), equivalent to 28 USD per mutation. For the larger c-SRC tyrosine kinase, the wall time using the same hardware and simulation conditions extended to 9 hours, resulting in 63 USD per mutation[10]. In the latter case, typical for a protein of biochemical or pharmacological interest, the wall time and computational cost are reduced by factors of up to 3 and 6, respectively, even if the sampling is concomitantly increased by a factor of 4. This translates into significant time and economic savings, while maintaining and even optimizing state-of-the-art accuracy and increasing precision, making QresFEP-2 an ideal protocol for high-throughput in silico mutagenesis studies.

## Domain-wide comprehensive mutagenesis

At this stage, the QresFEP-2 methodology has demonstrated satisfactory performance compared to other FEP methods like PMX or FEP+. However, the datasets analyzed so far are biased towards alanine mutations, which account for over 50% of the data points[45], and the vast majority of mutations (87.4%) involve substitutions with smaller side chains. To further validate its applicability in a broader context, we evaluated the performance of QresFEP-2 on a protein system subjected to comprehensive domain-wide mutagenesis[26]. In their original study, the Mayo group experimentally determined the thermodynamic stability of nearly every possible mutation in the small 56-residue B1 domain of streptococcal protein G (Gβ1). This provided a quintessential benchmark dataset for protein stability prediction tools, with the additional advantage, over previous datasets, of offering uniform data acquired from a single experiment[26]. Herein, we compared the FEP-calculated shifts in protein stability with the corresponding experimental data from this domain-wide assay. The mutational matrix considered in this part of the study comprised 456 data points, corresponding to data extracted for 38 positions × 12 side-chain mutations selected with the following considerations (see Fig. 6): the experimental data excluded both mutations on W43 and mutations to Trp, to avoid interferences with the Trp-based fluorescence assay; likewise, mutations to Cys were excluded to avoid oligomerization via disulfide formation; QresFEP-2 does not handle Pro (due to already discussed limitations of FEP methodologies), while mutations to/from titratable residues were initially excluded from this study to avoid the large fluctuations typical from change-changing mutations[10,41]; finally, terminal residues were avoided due to inconsistent comparisons with the tripeptide reference state.

Table 5 collects the statistics obtained from the QresFEP-2 calculations performed on the whole set of 456 mutations, the results shown in detail in Supplementary Table 12 and Fig. 7A. When analyzing these data, it is important to note a caveat regarding the experimental assay: while most data points have quantitative measurements of the experimental thermal shift (and thus an associated ΔΔG value), a subset of 57 data points (12.5% of the

**Fig. 6 | Design of the systematic mutation scan of Gβ1.** Amino acid sequence of the 56-residue B1 domain of streptococcal protein G, with the box color-coding indicating secondary structure elements. The outline of the boxes indicates the environment of the residue, and the residue single-letter color-coding visualizes side-chain category. Every position included was independently mutated to each of the remaining 12 side chains shown in the wheel chart on the right (excluding self-mutations), following the same residue single-letter color-coding.

**Table 5 | Results of the domain-wide comprehensive mutagenesis dataset (Gβ1), and for the dataset of 10 protein systems (Benchmark)**

| Dataset | $n$ | MAE (kcal · mol$^{-1}$) | Accuracy (%) | MCC | $R^2$ | $\rho$ | $\tau$ |
|---|---|---|---|---|---|---|---|
| Gβ1 | 399[a] | 1.27 $^{1.39}_{1.16}$ | 60.0 | 0.22 $^{0.32}_{0.13}$ | 0.30 $^{0.38}_{0.22}$ | 0.48 $^{0.57}_{0.40}$ | 0.34 $^{0.40}_{0.27}$ |
| | (456[b]) | — | (64.5) | (0.27 $^{0.36}_{0.18}$) | - | - | - |
| Benchmark (Table 3) | 583 | 1.25 $^{1.36}_{1.14}$ | 87.1 | 0.41 $^{0.52}_{0.30}$ | 0.49 $^{0.57}_{0.41}$ | 0.71 $^{0.76}_{0.67}$ | 0.53 $^{0.57}_{0.49}$ |
| Total (benchmark + Gβ1) | 982 | 1.26 $^{1.34}_{1.18}$ | 76.1 | 0.33 $^{0.40}_{0.27}$ | 0.47 $^{0.53}_{0.41}$ | 0.66 $^{0.70}_{0.62}$ | 0.48 $^{0.52}_{0.45}$ |

Accuracy expressed as % of correct predictions.

$n$ number of mutations, $R^2$ coefficient of determination, MAE mean absolute error, $\rho$ Spearman's rank correlation coefficient, $\tau$ Kendall rank correlation coefficient, MCC Matthews correlation coefficient.

[a]Includes only experimental quantitative values.

[b]Includes additional qualitative dataset ($n$ = 57). For these datapoints, only qualitative statistical figures of merit (i.e., accuracy and MCC) are meaningful and indicated in parenthesis.

total) corresponds to mutations leading to totally unstable protein, a folding Intermediate, or simply no expression, and were referred to in the original study as "qualitative dataset"[26]. With the exception of L5N and V54N (predicted to be stabilizing), our simulations of all mutants of this qualitative dataset resulted in significant destabilization or, in three cases (G41N, G41H, and G41V), unphysical models of the mutant side chain that include atomic clashes and led to simulation crashes, yielding a 96.5% true positive accuracy. While the entire dataset could be aggregated for a global analysis in terms of binary accuracy, yielding an MCC = 0.27, only the 399 data points with measured ΔΔG values were suitable for further quantitative analysis (Table 5).

A particularity of the quantitative dataset of 399 mutants is that, by excluding the mutations with a more pronounced effect on protein stability, the dataset is compressed towards zero, with the immediate consequence of overrepresenting mutations with neutral effect on stability. This is in clear contrast to the wider distribution of the benchmark dataset discussed above, as it can be clearly appreciated in Fig. 7B. The accuracy of QresFEP-2 on the quantitative dataset of Gβ1 domain reveals a MAE$_{Gβ1}$ = 1.27 kcal·mol$^{-1}$, consistent with the averaged result obtained from the previous benchmark (MAE$_{benchmark}$ = 1.25 kcal·mol$^{-1}$). However, the correlation coefficient is significantly lower ($R^2_{Gβ1}$ = 0.30, compared to $R^2_{benchmark}$ = 0.49, Table 5), which is expected given the narrower distribution of experimental data (see Fig. 7B). In other words, we are putting the lens on the mutations with moderate effects on protein stability.

This comprehensive and homogeneous dataset provides a valuable test set not only for evaluating the performance of QresFEP-2, but in principle also for trying to discern specific amino acid properties or local environments that may contribute to prediction inaccuracies. An initial analysis of the MAE for each of the 38 positions (averaged over the 12 mutations per position) reveals relatively consistent QresFEP-2 performance. Most position-averaged MAE values are under 2.0 kcal·mol$^{-1}$, with the only

exception of three threonine residues: T49, T51 and especially T25, which shows a larger average error of 3.4 kcal·mol$^{-1}$. Threonine is the most abundant side chain in this test set, and appears frequently solvent exposed in the Gβ1 structure, in some cases making polar interaction with neighboring residues that seem to be overestimated for these three residues; in other words, the reason of the lower performance of QresFEP-2 at these positions is a combination of the intrinsic nature of the side chain with the microenvironment. Interestingly, a heat map of the average results obtained per position projected on the 3D structure of the protein indicates good agreement with the experimental data (Fig. 8), demonstrating the ability of QresFEP-2 to detect regions where mutations will have stabilizing or destabilizing effects[26].

Dissection of the data by wild-type and mutant residue types, however, does not reveal any other clear patterns (Supplementary Table 13). The MAEs range from 0.81 kcal·mol$^{-1}$ for glutamine (averaged over 12 mutations) to 1.47 kcal·mol$^{-1}$ for glycine (averaged over 48 mutations). The unequal representation of different amino acids in the Gβ1 sequence, particularly the complete absence of histidine and serine, complicates making any conclusion in this sense. Similarly, while the frequency of mutant residues is more evenly distributed, with 35 datapoints on average per side chain, no clear patterns emerge, with MAEs for mutant residues ranging from 1.04 kcal·mol$^{-1}$ for asparagine to 1.61 kcal·mol$^{-1}$ for tyrosine.

## Site-directed mutagenesis and ligand binding

Another compelling application of FEP simulations of protein point mutations is the estimation of mutational effects on ligand binding. The single-topology annihilation protocol implemented in QresFEP-1 has proven extremely useful as a computational counterpart of experimental site-directed mutagenesis studies. It has served both to explain[22–24] and to design experiments, aimed at elucidating ligand binding modes on GPCRs[46,47]. To assess the performance of QresFEP-2 in this applicability domain, we selected the dataset

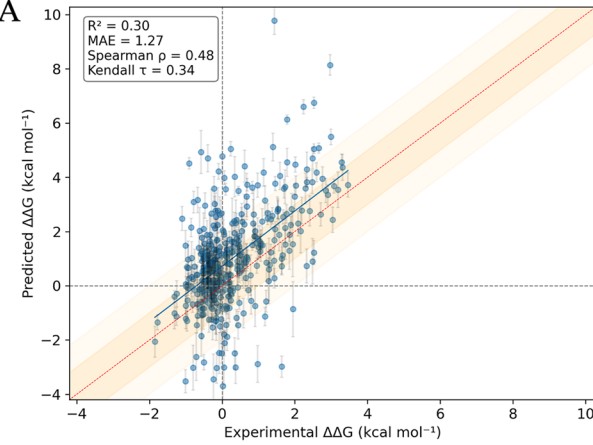
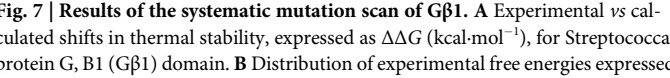
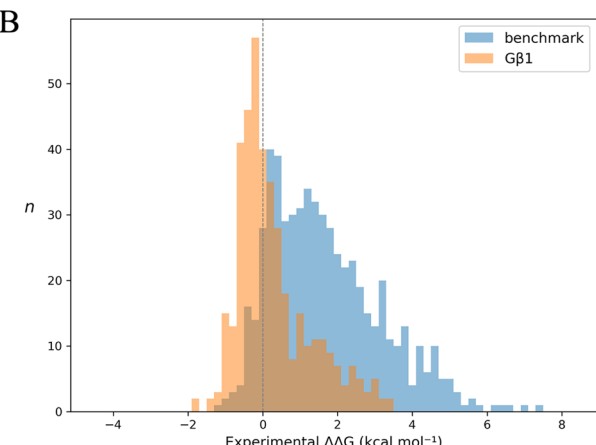

**Fig. 7 | Results of the systematic mutation scan of Gβ1. A** Experimental *vs* calculated shifts in thermal stability, expressed as $\Delta\Delta G$ (kcal·mol$^{-1}$), for Streptococcal protein G, B1 (Gβ1) domain. **B** Distribution of experimental free energies expressed as $\Delta\Delta G$ (kcal·mol$^{-1}$) for the benchmark dataset of 10 protein systems (blue histogram) and Gβ1 benchmark data (orange histogram). $N$ = counts, bin width is 0.2 kcal·mol$^{-1}$.

**Fig. 8 | Positional sensitivity of Gβ1 to point mutations.** Positions are colored by the $\Delta\Delta G_{stability}$ value obtained as the average of the 12 possible mutations at each position. Side chains are shown as lines for residues with a destabilizing positional sensitivity (grey to red, average $\Delta\Delta G < 0$). Residues not considered for prediction are colored in grey ($\Delta\Delta G = 0$). For comparison, this map is shown using the original coloring scheme from the experimental study, with the same sidechains represented in lines[26].

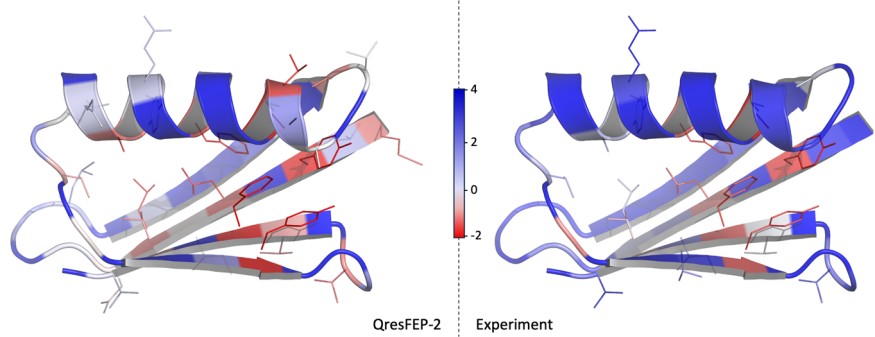

QresFEP-2 | Experiment

of 26 A$_{2A}$AR mutations on agonist binding previously characterized with QresFEP-1[22,23]. The thermodynamic cycle in this case involves performing the mutation in the folded protein environment in the presence (holo) or absence (apo) of the ligand of interest, in this case, the agonist NECA[22,23]. The results are presented in Fig. 9, Table 6, and Supplementary Table 12, where it can be appreciated that both methods exhibit a very similar accuracy in the predictions, with a slight improvement observed for QresFEP-2 being again statistically non-significant. The autocorrelation between both methods is shown in Supplementary Fig. 16. More remarkable is the computational efficiency of QresFEP-2, where similar convergence in the FEP-calculated binding affinity shift values (average SEM for the calculations is 0.9 kcal·mol$^{-1}$ in both cases), is achieved in a fraction of the calculation time, which is between 3 to 10-fold depending on the transformation type. This estimation of gain in computational efficiency was done considering the single thermodynamic cycle characteristic of the hybrid-topology approach (protocol H, Supplementary Fig. 2), as compared to the two independent thermodynamic cycles representative of successive annihilation to Ala from both *wt* and *mut* forms, needed for the 12 non-Ala mutations of this dataset, each of them requiring substantially longer sampling than the hybrid-topology protocol H here developed. Although this represents a limited dataset as compared to the previous protein-stability benchmark and Gβ1 case, the agonist-binding to A$_{2A}$AR mutational data covers a wide range of mutations, spread around the orthosteric binding site of the receptor and includes both direct and indirect contacts with the ligand (Fig. 9). Such a dataset constitutes thus a representative proof-of-concept of the applicability of the QresFEP-2 protocol on the ligand-binding affinity shifts induced by point mutations, including the characterization of site-directed mutagenesis or drug resistances, typical for pharmacological or clinical studies.

## Protein-protein interactions

Understanding protein-protein interactions (PPIs) at the molecular level is key to modulate cellular signaling or molecular complexes. PPIs constitute a target of increasing interest for the pharmaceutical industry, either by competing with one of the proteins, or by attempting to restore the effect of pathogenic mutations in the affinity between the proteins involved. To help advancing a detailed understanding of PPIs, the SKEMPI database collects binding free energy changes upon mutation for structurally resolved protein–protein interactions[48]. We decided to expand the validation of QresFEP-2 on a set of experimental mutational values of a PPI case extracted from the SKEMPI database. In particular, we selected the set of single-point mutations affecting the PPI between ribonuclease barnase (included in the general benchmark) and its protein inhibitor barstar. The resulting dataset consists of 11 neutral single-point mutations, located within the PPI interface. From these, six are mutations located on three positions of barnase while the remaining five mutations are located on four positions of barstar (Fig. 10A). This relatively small dataset covers however a variety of side chains, with mutations from H, Y, Q, T to A, F, Q, G, L (Fig. 10). QresFEP-2 was easily adapted to the PPI problem by designing a thermodynamic cycle where the mutation on either monomer is simulated in the presence or absence of the other monomer, to easily estimate the relative binding free energy difference between the *wt* and *mut* protein-protein complex. The results, shown in Table 7, Supplementary Table 13 and Fig. 10B, are encouraging and represent a proof-of-concept of the applicability domain of QresFEP-2 on modeling PPIs. Indeed, the corresponding statistical figures of merit are within the same values as reported for several of the protein systems benchmarked for thermal stability. The MAE of 1.64 kcal·mol$^{-1}$ is mostly affected by one outlier, representing the overprediction of the H102G

mutation in barstar, while the overal correlation with experimental values is excellent ($R^2$ = 0.73, see Table 7, Fig. 10B). A more detailed application of QresFEP-2 on PPIs is undergoing in our lab.

## Concluding remarks

Accurate prediction of mutational effects on protein stability or ligand binding presents significant challenges for computational simulations. One commonly recognized limitation in the field is the accuracy of protein force field parameters[49], but this is not the only or the most important factor. Appropriate modeling of the mutant side chain is crucial for the accuracy of the associated FEP prediction. Our approach for mutant modeling demonstrates a good balance between accuracy and computational efficiency, where the extensive MD sampling along the FEP transformation should account for potential structural rearrangements in the associated microenvironment. In this sense, the hybrid topology transformation, combined with the inclusion of a soft-core potential, ensures good

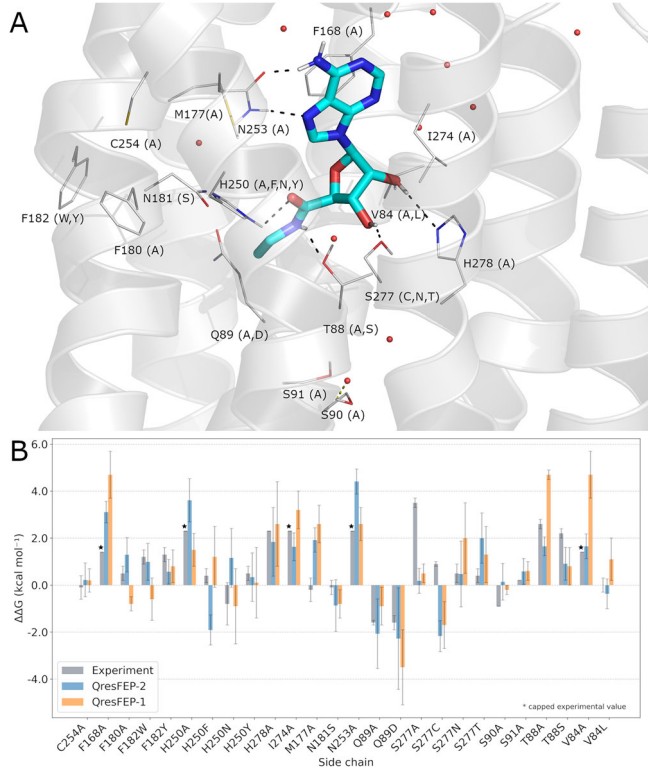

**Fig. 9 | Site directed mutagenesis study of the A$_{2A}$AR with QresFEP. A** A$_{2A}$AR (gray) in complex with NECA (cyan, PDB 2YDV). Residues undergoing mutation are depicted in lines, with the mutant version(s) in parenthesis. Experimental water molecules preserved in the simulation in red spheres. **B** Calculated (blue, QresFEP-2; orange, QresFEP-1) and experimental (gray) NECA binding free energy differences between each A$_{2A}$AR mutant and the *wt* receptor. The star symbol in the plot denotes that an experimental value could not be determined and represents the detection threshold in the corresponding experiment (as detailed in refs. 22,23).

convergence allowing for proper sampling of major conformational changes. The optimization of the sampling protocol on the T4L dataset shows an optimal balance between sufficient sampling (20 ns MD simulation) at a low computational cost (3 h wall time per mutation using a 160 CPU cluster), and high accuracy (MAE < 1.5 kcal·mol$^{-1}$). As compared to the commercial software FEP+, our protocol exhibits slightly improved accuracy (see Table 4 and Supplementary Fig. 15) obtained at a fraction of the computational cost, offering a competitive open-source alternative for large-scale FEP estimation of point mutation effects on protein stability.

However, there are situations where the initial configuration of the mutant might not be properly modeled, for instance, if it leads to the creation of a cavity and subsequent rearrangement of solvent molecules, and in such cases, the associated MD sampling may not fully overcome this limitation[50,51]. Elucidating whether this or other technical reasons (e.g., long-range effects of the mutation, impact on the protein folding process) underpin the behavior of outliers is not trivial from a purely structural or even energetic perspective. The lack of discernible systematic patterns necessitates individual analysis of each mutation to understand the origins of prediction errors. Akin to finding a needle in a haystack, this task of such detailed analysis poses a laborious challenge for human efforts, while it is essential for improving prediction accuracy. However, rapid advancements in artificial intelligence (AI) and machine learning offer promising avenues for automating this process. By defining appropriate features that capture the relevant chemical and physical properties of the protein, water environment, and wild-type and mutant amino acids, a neural network model could potentially learn to predict which mutations are likely to yield accurate or inaccurate stability predictions. In future work, we aim to leverage AI-driven approaches in conjunction with physics-based methods for enhanced protein stability prediction.

QresFEP-2 proves to be a physics-based versatile method to evaluate various effects of protein mutations based on estimations of the associated free energy changes. The method is a hybrid-topology evolution of its predecesor QresFEP, originally developed as a single-topology alanine-scan protocol adapted to non-alanine mutations by relatively doubling the computational cost. QresFEP-2 is here initially benchmarked on 10 protein systems, accounting for almost 600 mutations from which ~50% are non-alanine mutations, and further tested on a comprehensive domain-wide mutagenesis dataset containing 400 mutations of evenly distributed nature (Fig. 6). While all these benchmark and test datasets consist of mutations affecting protein stability, we also demonstrate the applicability to other biochemical phenomena of interest, namely protein-ligand binding affinity shifts induced by point mutations and protein-protein interactions. In both areas, the results showcase a promising trade-off between the accuracy and high scalability of QresFEP-2, rendering this method attractive for routine evaluation of mutational effects on ligand binding, common in pharmaceutical drug design projects.

## Methods

### QresFEP-2 API

QresFEP-2 comprises a collection of Python scripts, readily installable on any operating system using the provided Conda environment, and freely accessible on GitHub [https://github.com/qusers/qligfep]. It offers a robust and efficient pipeline for setting up and analyzing FEP simulations of amino

### Table 6 | Site-directed mutagenesis results for the A$_{2A}$AR-NECA system

| Method | n | MAE (kcal · mol$^{-1}$) | Accuracy (%) | MCC | $R^2$ | $\rho$ | $\tau$ |
|---|---|---|---|---|---|---|---|
| QresFEP-2 | 26 | 1.12 $^{1.47}_{0.79}$ | 76.9 | 0.43 $^{0.78}_{-0.02}$ | 0.31 $^{0.63}_{0.07}$ | 0.53 $^{0.81}_{0.16}$ | 0.41 $^{0.65}_{0.11}$ |
| QresFEP-1 | 26 | 1.30 $^{1.69}_{0.95}$ | 76.9 | 0.46 $^{0.80}_{0.04}$ | 0.34 $^{0.64}_{0.09}$ | 0.60 $^{0.82}_{0.25}$ | 0.45 $^{0.69}_{0.17}$ |

Accuracy expressed as % of correct predictions.

*n* number of mutations, $R^2$ coefficient of determination, *MAE* mean absolute error, $\rho$ Spearman's rank correlation coefficient, $\tau$ Kendall rank correlation coefficient, *MCC* Matthews correlation coefficient.

acid mutations within the Q molecular dynamics (MD) software, which is specifically designed for various types of free energy simulations[25].

MD simulations in Q are typically performed under spherical boundary conditions, employing the Surface Constrained All-atom Solvent Model (SCAAS) in conjunction with the local reaction field (LRF) method for evaluating long-range electrostatic interactions[33,52]. This setup effectively reduces the computational cost compared to the more popular periodic boundary conditions (PBC) that encompass the entire biomolecule and its periodic images. By focusing on a region of interest, typically a 50 Å diameter sphere centered on the residue undergoing mutation, this approach maintains the accuracy of the free energy simulation, as we have previously demonstrated[53].

For each mutation, the complete FEP pathway consists of two subperturbations in which atomic charges are gradually annihilated/created, and Van der Waals parameters transition through a soft-core stage. Each subperturbation is divided into a number of λ-windows, distributed either evenly (i.e., linear λ-sampling) or with a higher density of windows near the end points (i.e., sigmoidal λ-sampling). The free energy change (ΔΔG) between the two end-states of the subperturbation can be estimated using Zwanzig's exponential equation[54]:

$$\Delta\Delta G = \Delta G_B - \Delta G_A = -\beta^{-1} \sum_{i=1}^{n-1} ln \left\{ e^{-\beta\left(U_{i+1}-U_i\right)} \right\}_A \qquad (1)$$

where $\beta = 1/kT$, $U_i$ represents the effective potential energy function of a specific FEP window λ, and $n$ is the number of intermediate λ-states. $U_i$ is constructed as a linear combination of the initial () and final ($B$) potentials of the subperturbation

$$U_i = U_A + \lambda_i(U_B - U_A) \qquad (2)$$

where the coupling parameter λ is incrementally increased from 0 to 1 in $n$ discrete steps. Alternatively, the free energy difference between two adjacent windows ($\Delta\Delta G_i$) may also be estimated using Bennett's acceptance ratio (BAR) method[55]:

$$\Delta G_i = -\beta^{-1} ln \frac{\left\langle 1 + e^{-\beta\left(\Delta U \Delta\lambda_i - C_i\right)} \right\rangle_{i+1}}{\left\langle 1 + e^{+\beta\left(\Delta U \Delta\lambda_i - C_i\right)} \right\rangle_i} + C_i \qquad (3)$$

where the constants $C_i$ are iteratively optimized so that the two ensemble averages become equal, yielding $\Delta G_i = C_i$. With either Zwanzig or BAR estimations, the concatenation of the free energy differences of the two successive subperturbations provides the calculated free energy of a mutation within a specific environment (e.g., vacuum, aqueous solution, protein).

The QresFEP-2 protocol parameters optimized in this study (see Supplementary Fig. S2) consist of 50 unevenly spaced (sigmoidal) λ-windows for each of the 2 FEP stages, with 20 ps of MD sampling per λ-window. These parameters are easily changed by the user, but it is essential that the same FEP protocol is applied to both the protein and the reference tripeptide systems, enabling the calculation of relative protein stability free energies by solving the corresponding thermodynamic cycle (Fig. 3)[41]. Throughout this process, the pairwise non-bonded interactions between the side-chain atoms of the *wt* and *mut* residues are excluded from the calculations. Furthermore, the bonded terms theoretically connecting the two side chains through the common Cα are deactivated (the Cβ$_{wt}$–Cα–Cβ$_{mut}$ bond angle and the x–Cβ$_{wt}$–Cα–Cβ$_{mut}$ and Cβ$_{wt}$–Cα–Cβ$_{mut}$–y torsions, where x and y represent any other atom bound to the Cβs). This ensures that the two sets of atoms do not "feel" each other and that the system is not artificially influenced by unphysical connections. The disappearing atoms gradually transition to dummy atoms, which only interact through bonded terms[30].

For the side-chain mimics, relative hydration free energies were calculated from simulations in water and vacuum spheres. In all cases, the FEP result of each leg of the thermodynamic cycle is obtained as an average over 10 independent replicate MD simulations with identical parameters (varying random initial velocities sampled from a Maxwell-Boltzmann distribution), and the associated standard error of the mean (SEM) estimated in each case.

**A**

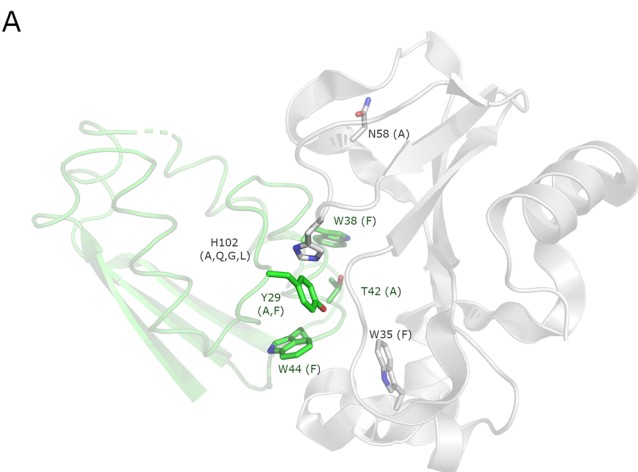

**B**

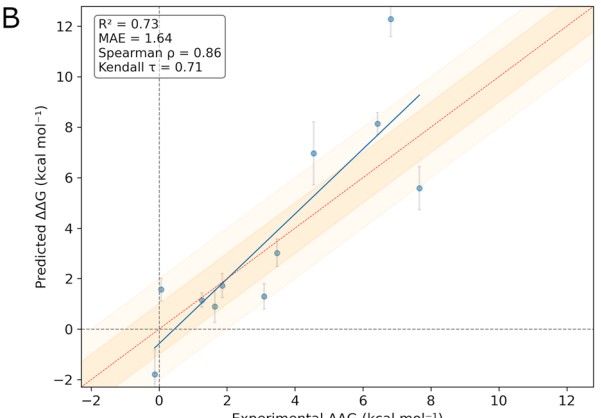

**Fig. 10 | Point mutations affecting the PPI barnase-barstar. A** Ribonuclease barnase (gray) in complex with Barstar (green, PDB 1BRS). Residues undergoing mutation are depicted in lines, with the mutant version(s) in parenthesis. **B** Experimental *vs* calculated shifts in barnase-barstar binding affinity, expressed as ΔΔG (kcal·mol⁻¹), for the 11 mutants considered (Table 7).

**Table 7 | Results of QresFEP2 for the protein–protein interaction barnase–barstar**

| Complex | PDB ID | n | MAE (kcal · mol⁻¹) | Accuracy (%) | MCC | $R^2$ | $\rho$ | $\tau$ |
|---|---|---|---|---|---|---|---|---|
| Barnase–Barstar | 1BRS | 11 | 1.64 $^{2.59}_{0.91}$ | 100 | 1.00 $^{1.00}_{1.00}$ | 0.73 $^{0.94}_{0.54}$ | 0.86 $^{1.00}_{0.53}$ | 0.71 $^{1.00}_{0.33}$ |

Accuracy expressed as % of correct predictions.

*n* number of mutations, *$R^2$* coefficient of determination, *MAE* mean absolute error, *ρ* Spearman's rank correlation coefficient, *τ* Kendall rank correlation coefficient, *MCC* Matthews correlation coefficient.

## System preparation

The structural models for each protein system were derived from refined crystal structures, with PDB entries listed in Table 4. Structure preparation was performed using Schrödinger's Maestro (Schrödinger Suite Release 2021-1, v12.7.161)[56], accounting for necessary asparagine and glutamine flips, and assigning histidine protonation states at pH 7.0. The protonation states of titratable residues were assigned using PropKa (v3.1)[57]. Non-protein heterogroups were removed, and only water molecules with at least one direct hydrogen bond to a protein atom were retained. Additional modifications were needed for structure 1TEN, where terminal residue Arg802 was removed due to missing backbone atoms, and for structure 1FMK, where phosphorylated tyrosine Ptr527 was dephosphorylated. The missing loop region between Arg409 and Phe424 in 1FMK was modeled with Prime[56], as well as missing side-chain atoms for Met59 in 1YPC and Phe424 in 1FMK.

The structural model used for the $A_{2A}$-NECA complex was prepared in analogy to our previous work from ref. 22,23. Briefly, the active-like $A_{2A}$AR structure with PDB code 2YDV was refined with Protein Preparation Wizard in Maestro[56], inserted in the membrane and equilibrated under PBC with the GPCR-ModSim protocol[58]. Ligand parameters from the OPLSAA force field were retrieved from Schrödinger's ffld module[56], and translated into Q with the automated protocol implemented in QresFEP-2. Experimental relative binding free energies ($\Delta\Delta G_{exp}$) calculated from $K_i$ reported values as:

$$\Delta\Delta G_{bind}^{exp} = RT\ln(K_i^{mut}/K_i^{wt}). \tag{4}$$

The initial conformations for mutant residues were generated with PyMOL mutagenesis tool (v3.0)[43], selecting the most probable side-chain rotamers in all cases. For comparison, complete structural models of each single-point mutant were generated using AlphaFold2[12]. The mutant residue was then extracted from the generated structure for subsequent topology building. Tripeptide structures representing the unfolded state were also generated with PyMOL, where the protein was truncated on either side of the mutable residue's flanking residues, while capping the flanking residues. For simulations of hydration free energies of side-chain mimics, initial 3D configuration of the side chain was obtained with PyMOL and the Cα atom was replaced with a hydrogen.

## Molecular dynamics

Molecular dynamics (MD) simulations were performed using the Q software package (v6.0)[59], employing the OPLS-AA/M force field for proteins and the TIP3P water model[35]. Systems were solvated in a spherical water droplet with a diameter of 50 Å, centered on the Cβ atom (or the side-chain hydrogen in the case of glycine) of the amino acid undergoing the FEP transformation. All atoms inside the simulation sphere were allowed to move freely, while protein atoms outside the sphere were tightly harmonically constrained to their initial coordinates with a force constant of 200 kcal·mol⁻¹·Å⁻² and excluded from non-bonded interactions. Water molecules at the sphere boundary were subjected to radial and polarization restraints according to the SCAAS model to mimic bulk water properties[33]. Ionizable residues within the sphere outer layer (<3 Å from the surface) were neutralized to prevent artifacts arising due to insufficient dielectric screening. All non-bonded interactions involving atoms in the transforming amino acids (so-called Q atoms) were calculated explicitly within the sphere. For all other atoms, Lennard-Jones interactions were truncated beyond 10 Å, and long-range electrostatic interactions beyond this cutoff were treated with the LRF multipole expansion method[52]. Protein and tripeptide simulations used a 2 fs time step, enabled by the use of the SHAKE algorithm to constrain bonds involving hydrogens and solute bonds and angles, whereas simulations of the side-chain mimics used a 1 fs time step for comparative purposes with previous versions. Each simulation was run with 10 independent replicas, initiated with different random velocities. The simulation protocol included an initial structural optimization for 10 ps at 0 K temperature, followed by gradual heating to 298 K temperature over

150 ps. Subsequently, harmonic restraints (10.0 kcal·mol⁻¹·Å⁻²) on solute heavy atoms were released over 350 ps while concurrently relaxing the thermostat bath coupling time from 0.2 to 10 fs. This was followed by an unrestrained 0.5 ns equilibration period at 298 K. However, topologically equivalent heavy atoms in the *wt* and *mut* side chains were subjected to harmonic distance restraints (10.0 kcal/mol/Å²) according to the automated dynamic restraining scheme throughout the simulations. The FEP/MD production phase, during which energy averages were collected, involved (using default protocol H) 2 ns of simulation time per replica, totaling 20 ns per leg of the thermodynamic cycle and 40 ns for the complete FEP cycle.

## Data availability

The authors declare that the data supporting the findings of this study are available within the paper and its Supplementary Information files. Should any raw data files be needed in another format they are available from the corresponding author upon reasonable request. Source data are provided with this paper under "Supplementary Data 1".

## Code availability

The QresFEP-2 code is freely available under the following community repository: https://github.com/qusers/qligfep (https://doi.org/10.5281/zenodo.8312554).

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

## Acknowledgements
Support from the Swedish Research Council (grant no. 2022-03441) and the Knut and Alice Wallenberg Foundation (grant no. 2023.0210) is gratefully acknowledged. This study was supported by grant PID2023- 150793OB-I00 from the Spanish Ministry of Science and Innovation – State Research Agency – FEDER-UE, and is part of the project Novel Oncological Targets – Inhibiting Cancer via Mutated G Proteins with file number VI.Veni.232.243 (partly) financed by the Dutch Research Council (NWO). Computational resources were provided by the National Academic Infrastructure for Supercomputing in Sweden (NAISS) partially funded by the Swedish Research Council through grant agreements no. 2022-06725 and no. 2018-05973.

## Author contributions
L.K. and N.V.D.B. performed the experiments. H.G.T., W.J., J.Å and L.K. designed the study. L.K. and H.G.T. analyzed the data. L.K., W.J., J.Å and H.G.T. wrote the paper.

## Funding

## Competing interests
The authors declare no competing interests.
