## [Transparent Peer Review file · Communications Chemistry]

Accurate Predictions of Protein Mutational Effects Accelerated with a Hybrid-Topology Free Energy Protocol

Corresponding Author: Professor Hugo Gutiérrez-de-Terán

Version 0:

Reviewer comments:

Reviewer #1

(Remarks to the Author)

This manuscript describes a new computational method for the prediction of the effects of mutations on protein stability. It is based on the previous method from co-authors using FEP computations (QresFEP), with some novel aspects such as the use of hybrid topology to perform FEP calculations. The field is well described in the Introduction, clearly stating the limitations of state-of-the-art methods. In this time of AI-based applications, it is encouraging to have studies using physics-based approaches, with the purpose of having a good mechanistic understanding of molecular behaviour, in addition to better predictors. While most of the methodology and approaches are well explained, there are missing details for some essential parts of the developed methods, including validation and comparison with previous approaches, in order to better evaluate the novelty and improvement with respect to other methods. Below are my detailed comments:

Major comments:

- The concepts of single-, hybrid- and dual-topology are essential to understand the novelty, and they are minimally defined in Figure 1 legend, but for the sake of clarity, it would be important to further discuss these concepts in the manuscript.
- Table 1 shows the computed solvation values relative to methane, and they are compared to the experimental values. It is not clear whether the experimental values are also relative to methane. Otherwise, it would be interesting to show the computed values relative to other solvents.
- Since they that the methodology is suitable for a range of applications, including protein-protein interactions (PPIs), it would be interesting to show its validation on a set of experimental mutational values from SKEMPI database affecting PPIs.
- The description of the different protocols (A, B, C....) is confusing. They are not explained in detail, and it is not clear the rationale behind all these different protocols, or the interpretation of their results. Defining multiple conditions for a predictor and choosing the best one can be seen as biased if the selection is not based on robust criteria. One concern is that this could lead to predictors that might not be of general applicability beyond current validation examples. Comparison with the previous QresFEP version (not just with the commercial FEP+ software) should be shown for all data sets, in order to evaluate better the improvement of predicting capabilities of this new proposed methodology.
- AF2 is used to model the mutations, which showed no significant difference with respect to using PyMol. It would be also interesting to try the most updated AF version 3.

Minor comments:

- There is a typo in Figure 3, missing a delta in the bottom equation (before G_{U_FEP}).

Reviewer #2

(Remarks to the Author)

The manuscript presents QresFEP-2, an automated, physics-based approach designed to estimate relative free energy changes due to single-point mutations in proteins. The authors describe a hybrid topology method for molecular dynamics (MD) sampling in free energy perturbation (FEP) calculations, with a focus on avoiding changes in atom types or bonded parameters to ensure both accuracy and automation. The work is compared against previous methods like QresFEP-1, FEP+, and PMX, with experimental validation using various datasets (e.g., T4 lysozyme, Barnase). The paper claims

substantial improvements in computational efficiency and accuracy over previous protocols.

1. While the hybrid topology approach may indeed be more efficient than previous single- or dual-topology approaches, the manuscript fails to critically address potential limitations of this method. For example, how does the hybrid approach impact the flexibility of side-chain interactions during FEP? Are there cases where this method might fail to model key interactions accurately, especially when the side-chain has a significant role in the protein's overall conformation?

2. The dynamic restraint approach seems promising, but there is no in-depth discussion about the scenarios where the imposed restraints might interfere with the natural conformational freedom of the system. Is there a risk of over-restraining in specific mutations, or could this impact the results in more flexible protein systems?

3. The authors claim that no changes in atom types or bonded parameters occur with the hybrid topology approach. However, it would be important to discuss in greater depth how this might affect non-covalent interactions, especially in larger systems with significant structural changes. Does the lack of adjustment in bonded parameters introduce systematic errors that might lead to inaccurate free energy predictions?

4. While the manuscript provides some data on solvation free energies (Table 1), the discussion surrounding this section is somewhat superficial. How do these results compare to other published methods beyond the immediate comparison to QresFEP-1 and QligFEP? How do the authors justify that the QresFEP-2 method consistently outperforms these alternatives, and are there specific scenarios where QresFEP-2 might be expected to underperform?

5. Table 2 presents the results for the T4 lysozyme alanine scan, but the manuscript doesn't adequately explain how the computational efficiency (reduction in sampling time) influences the overall accuracy of the free energy predictions. Furthermore, while the manuscript claims improvements in the efficiency of the QresFEP-2 protocol (compared to FEP+ and QresFEP-1), there is no detailed statistical comparison (e.g., p-values or confidence intervals) to fully support these claims. The conclusion that QresFEP-2 reduces computational time by 2 to 4-fold is significant, but it is not accompanied by an appropriate validation of this claim.

6. The manuscript often refers to performance improvements relative to FEP+ and PMX (e.g., in protein stability prediction), but the statistical analysis provided is weak. For example, the reported "non-significant improvements" between different protocols should be followed by more rigorous statistical tests to quantify these differences. Without such analysis, the comparison between methods feels superficial and insufficient.

7. There are also no mentions of other recent state-of-the-art methods or software packages beyond FEP+, QresFEP-1, and PMX, such as those incorporating newer force fields or improved solvation models. This limits the broader applicability of the claims made for QresFEP-2.

8. The manuscript primarily relies on T4 lysozyme and Barnase datasets for validation. While these are well-established systems, they do not fully represent the diversity of protein types and mutation scenarios that might arise in more complex biological systems. More diverse datasets (e.g., membrane proteins, multi-subunit complexes) should be included to evaluate the robustness of QresFEP-2 under different conditions.

9. The manuscript evaluates the method with a subset of alanine mutations and a limited number of non-alanine mutations. For a method claiming broad applicability, it is crucial to include a wider range of mutations (e.g., large-scale deletions, insertions, and post-translational modifications) to demonstrate generalization.

10. In several instances, the authors claim "improvements" in accuracy (e.g., in Table 2, where QresFEP-2 slightly outperforms QresFEP-1 and FEP+). These improvements are typically within a small margin of error, and the lack of significant statistical differentiation between methods raises the question of whether such improvements are practically meaningful or if they fall within the typical experimental uncertainty.

Reviewer #3

(Remarks to the Author)

Koenekoop et al. presented QresFEP-2, a hybrid-topology free energy perturbation (FEP) protocol designed for accurate and efficient energetic assessment of protein mutations. The hybrid-topology approach offers advantages over both the single-topology method used in QresFEP-1 and the dual-topology method employed in FEP approaches like FEP+. The authors demonstrated the accuracy, robustness, and efficiency of QresFEP-2 through extensive benchmarking and validation, including: (1) solvation free energy calculations of amino acid side-chain mimics, (2) an alanine scan on the T4L dataset (n = 43), (3) alanine and non-alanine scans on the T4L (n = 66) and Barnase (n = 109) datasets, (4) a broad benchmark across 10 protein systems (n = 583), (5) domain-wide mutagenesis assessment on G β 1 (n = 399), and (6) site-directed mutagenesis on A2AAR (n = 26).

Overall, this is an excellent study, and the QresFEP-2 protocol is likely to be widely adopted due to its high accuracy and improved speed relative to FEP+.

However, I do have some concerns regarding the manuscript's writing and organization, which currently detract from the readability. Notably, there is no Table 3, yet there are two instances each of Table 4 and Table 5, which is quite confusing. I strongly encourage the authors to carefully review and correct these inconsistencies.

Version 1:

Reviewer comments:

Reviewer #1

(Remarks to the Author)

All major concerns have been satisfactorily addressed.

Reviewer #2

(Remarks to the Author)

The authors have provided satisfactory and scientifically sound responses to the concerns raised, and the manuscript appears to have been strengthened as a result of the revisions.

Reviewer #1: (...) *While most of the methodology and approaches are well explained, there are missing details for some essential parts of the developed methods, including validation and comparison with previous approaches, in order to better evaluate the novelty and improvement with respect to other methods. Below are my detailed comments:*

1. The concepts of single-, hybrid- and dual-topology are essential to understand the novelty, and they are minimally defined in Figure 1 legend, but for the sake of clarity, it would be important to further discuss these concepts in the manuscript.

RESPONSE: We agree that these concepts need a clear presentation. We have expanded the original description (provided in p 6-7 of the manuscript) of the different single-, dual- and hybrid-topology representations for residue sidechains.

2. Table 1 shows the computed solvation values relative to methane, and they are compared to the experimental values. It is not clear whether the experimental values are also relative to methane. Otherwise, it would be interesting to show the computed values relative to other solvents.

RESPONSE: The experimental values are also relative to methane, the value of which was also reported in Wolfenden et al. as representative of the sidechain of Alanine. This is now stated on p 11 and reflected on the title of Table 1. We also replaced “solvation” by “hydration” to avoid further confusion.

3. Since they that the methodology is suitable for a range of applications, including protein-protein interactions (PPIs), it would be interesting to show its validation on a set of experimental mutational values from SKEMPI database affecting PPIs.

RESPONSE: We appreciate this suggestion and provide in the revised version a proof of concept of the generalizability of our method to characterize PPIs. We apply the QresFEP-2 protocol to the 11 single-point mutations of the Barnase/Barstar complex, extracted from SKEMPI database (new ref 49). The results, reported in Fig 10, Table 7, Supplementary Table 15 and described in the new text in p 30, are very encouraging with the statistical parameters in similar order of magnitude as in the thermal-stability and protein-ligand binding cases. We have noted on the abstract and conclusions this proof-of-concept on the applicability domain of QresFEP-2 on PPIs. A more specific application of QresFEP-2 on PPIs is undergoing in our lab.

4. The description of the different protocols (A, B, C....) is confusing. They are not explained in detail, and it is not clear the rationale behind all these different protocols, or the interpretation of their results. Defining multiple conditions for a predictor and choosing the best one can be seen as biased if the selection is not based on robust criteria. One concern is that this could lead to predictors that might not be of general applicability beyond current validation examples. Comparison with the previous QresFEP version (not just with the commercial FEP+ software) should be shown for all data sets, in order to evaluate better the improvement of predicting capabilities of this new proposed methodology.

RESPONSE: We appreciate this comment, and have added on page 13 a brief description of the rationale behind the experimental design of the different MD-sampling protocols. On p 14 we also added a sentence about the general applicability of the proposed protocol, with a reminder of the flexibility of QresFEP-2 in setting up different sampling parameters.

A systematic comparison of QresFEP-2 with QresFEP-1 would be high consuming (i.e., we should produce QresFEP-1 calculations for almost 1.000 datapoints!) and not informative: we already have proven that the computational efficiency of QresFEP-2 is substantially superior to its predecessor (see that we further elaborate on response to comment 5 of referee #2), while the qualitative accuracy on both the solvation free energy and T4L alanine datasets is significant.

5. AF2 is used to model the mutations, which showed no significant difference with respect to using PyMol. It would be also interesting to try the most updated AF version 3

RESPONSE: Following this suggestion, we now added in Supplementary Figure 4 a column with the results of the corresponding FEP calculations performed on AF3 generated mutants.

On page 17 we compare the results of all three methods to generate mutants (PyMOL, AF2 and AF3), concluding that “*The results (see new Supplementary Fig. 4) clearly show no significant differences between the three sets of calculations, (...)*”

6. There is a typo in Figure 3, missing a delta in the bottom equation (before G_U_FEP).

RESPONSE: Thank you, corrected as suggested.

Reviewer #2 : (...) *The paper claims substantial improvements in computational efficiency and accuracy over previous protocols.*

1. While the hybrid topology approach may indeed be more efficient than previous single- or dual-topology approaches, the manuscript fails to critically address potential limitations of this method. For example, how does the hybrid approach impact the flexibility of side-chain interactions during FEP? Are there cases where this method might fail to model key interactions accurately, especially when the side-chain has a significant role in the protein's overall conformation?

RESPONSE: The proper treatment of sidechain flexibility is indeed a general concern valid for any FEP method, as we already discussed on p 17 in relation to the modeling of the initial conformation of the mutant sidechain. The particular implementation of our hybrid-topology approach is not hampering the exploration of flexibility, which mainly depends on the MD sampling (see Suppl Fig 2 and updated description in p 13-14, on request to comment 1 of referee #1).

To respond the second question, we remind the referee of paragraph 2 of the Conclusions, where we discuss that “*there are situations where the initial configuration of the mutant might not be properly modeled (...) Elucidating whether this or other technical reasons underpin the behavior of outliers is not trivial.*” And anticipate that “*In future work, we aim to leverage AI-driven approaches in conjunction with physics-based methods for enhanced protein stability prediction*”

2. The dynamic restraint approach seems promising, but there is no in-depth discussion about the scenarios where the imposed restraints might interfere with the natural conformational freedom of the system. Is there a risk of over-restraining in specific mutations, or could this impact the results in more flexible protein systems?

RESPONSE: We extensively illustrate this point in the Tyr → Phe test case transformation presented in Fig. 2 and Supplementary Movie 1, and keep the restraints to the minimum possible to prevent flapping. We remind that the dynamic restrains applied on each case strongly depend on the initial modeling of the mutant sidechain, and we refer to the previous point of this referee for a discussion on this sense.

3. The authors claim that no changes in atom types or bonded parameters occur with the hybrid topology approach. However, it would be important to discuss in greater depth how this might affect non-covalent interactions, especially in larger systems with significant structural changes. Does the lack of adjustment in bonded parameters introduce systematic errors that might lead to inaccurate free energy predictions?

RESPONSE: The concern is probably due to miscommunication from our side, solved by re-writing the corresponding sentence on p7 on the new version of the manuscript, which now reads: “*QresFEP-2 avoids transformation of atom types or any bonded parameters (i.e., the set of atoms with associated parameters representing mut gradually replaces the wt set of atoms and parameters)*”. Indeed, any FEP transformation following dual-like topology does not involve *transformation* of bonded parameters, but rather all atoms and associated parameters in topology 1 (wt) are gradually *replaced* by the corresponding atoms and associated parameters in topology 2 (mutant). This involves both non-bonded and bonded parameters, which are thus properly modeled according to the associated forcefield.

4. While the manuscript provides some data on solvation free energies (Table 1), the discussion surrounding this section is somewhat superficial. How do these results compare to other published methods beyond the immediate comparison to QresFEP-1 and QligFEP? How do the authors justify that the QresFEP-2 method consistently outperforms these alternatives, and are there specific scenarios where QresFEP-2 might be expected to underperform?

RESPONSE: The referee has probably missed that Supplementary Table 1 provided the solvation free energies of the same dataset calculated with different FEP and TI methods, and the corresponding description on the main text stated “*QresFEP-2 ranks as the second most accurate protocol*”. We now added in p 12 the following sentence to avoid the reader missing this point.: “*The performance of QresFEP-2 was compared to other published methods beyond QresFEP-1 (Supplementary Table 1).*”

5. Table 2 presents the results for the T4 lysozyme alanine scan, but the manuscript doesn't adequately explain how the computational efficiency (reduction in sampling time) influences the overall accuracy of the free energy predictions. Furthermore, while the manuscript claims improvements in the efficiency of the QresFEP-2 protocol (compared to FEP+ and QresFEP-1), there is no detailed statistical comparison (e.g., p-values or confidence intervals) to fully support these claims. The conclusion that QresFEP-2 reduces computational time by 2 to 4-fold is significant, but it is not accompanied by an appropriate validation of this claim.

RESPONSE: First, we want to clarify that we did NOT claim any statistical significance of the trends on T4L, as the original version of the manuscript stated in p 14 “*A slight, statistically non-significant improvement in the predictions (...)*”.

In any case, we appreciate the suggestion and have replaced in all Tables (1-7) the former statistical analysis (based on the bootstrapped SEM obtained for most statistical parameters) by confidence intervals (CIs) obtained at 95% by bootstrapping (also following the recommendations of new reference 38, Mey et al). We maintain the only claim of statistically significant improved accuracy of QresFEP-2 over FEP+, which referred to the overall dataset of 10 protein systems, 534 common datapoints. Following the advice of the referee, this claim is now backed up by a Mann-Whitney-Wilcoxon non-parametric test ($p < 0.0001$) (see new Fig S15).

The claim of a 2-4 fold reduction in computational time of QresFEP-2 over QresFEP-1 for alanine mutations is now illustrated on p 14 with a numerical example: “*As an example, mutation of a mid-size aminoacid (Ile \rightarrow Ala) with the single-topology gradual annihilation of QresFEP-1 involves 6 subperturbations \times 50 λ -windows \times 10,000 (1 fs) steps = 3M steps, while the same mutation in QresFEP-2 involves 2 stages \times 50 λ -windows \times 10,000 (2 fs) steps = 1M steps.*”

6. The manuscript often refers to performance improvements relative to FEP+ and PMX (e.g., in protein stability prediction), but the statistical analysis provided is weak. For example, the reported "non-significant improvements" between different protocols should be followed by more rigorous statistical tests to quantify these differences. Without such analysis, the comparison between methods feels superficial and insufficient.

RESPONSE: This comment is responded in the previous point. We have replaced the previous SEM values of the statistical figures of merit by the corresponding 95% CIs, and only claim statistical significance on the global benchmark dataset, on the basis of the statistical test therein described.

7. There are also no mentions of other recent state-of-the-art methods or software packages beyond FEP+, QresFEP-1, and PMX, such as those incorporating newer force fields or improved solvation models. This limits the broader applicability of the claims made for QresFEP-2.

RESPONSE: We have compared our method to the only existing FEP packages that, to the best of our knowledge, have been benchmarked for protein-stability effects of point mutations on at least one protein system from those initially considered for benchmarking: QresFEP-1

(T4L Ala mutations), PMX (barnase) and FEP+ (10 protein systems). We are not aware of other efforts in this sense.

8. The manuscript primarily relies on T4 lysozyme and Barnase datasets for validation. While these are well-established systems, they do not fully represent the diversity of protein types and mutation scenarios that might arise in more complex biological systems. More diverse datasets (e.g., membrane proteins, multi-subunit complexes) should be included to evaluate the robustness of QresFEP-2 under different conditions

RESPONSE: The referee is simply wrong in this point: QresFEP-2 is herein benchmarked on 10 protein systems (including T4L and Barnase), accounting for 583 initial datapoints. We further validate the applicability to thermal stability data with a comprehensive domain-wide mutagenesis dataset consisting of additional 456 datapoints. Thereafter, we evaluate applicability on two other biochemical problems: site-directed mutagenesis data on a GPCR (a membrane protein included in the initial version of the manuscript), and now also on the barnase/barstar system, included in this revised version following the suggestion in comment 3 of referee #1, to illustrate the applicability on PPIs.

9. The manuscript evaluates the method with a subset of alanine mutations and a limited number of non-alanine mutations. For a method claiming broad applicability, it is crucial to include a wider range of mutations (e.g., large-scale deletions, insertions, and post-translational modifications) to demonstrate generalization

RESPONSE: This is also incorrect: the initial benchmark comprises 583 mutations collected from 10 different protein systems, from which 50.2% are alanine mutations, the rest being non-alanine mutations. Further validation on a comprehensive domain-wide mutagenesis dataset contained just 7.3% of alanine while 92.7% were non-alanine mutations! We have added this analysis on the conclusions of the manuscript, to actually claim broad applicability of the method.

Evaluation of large-scale deletions, insertions, and post-translational modifications is out of the scope of any FEP-based method, which is obvious to anyone familiar with these methods.

10. In several instances, the authors claim "improvements" in accuracy (e.g., in Table 2, where QresFEP-2 slightly outperforms QresFEP-1 and FEP+). These improvements are typically within a small margin of error, and the lack of significant statistical differentiation between methods raises the question of whether such improvements are practically meaningful or if they fall within the typical experimental uncertainty.

RESPONSE: The referee seems to be insisting on the same topic as already raised on points 5 and 6 of this same referee. We insist on our response to those comments.

Reviewer #3: (...) Overall, this is an excellent study, and the QresFEP-2 protocol is likely to be widely adopted due to its high accuracy and improved speed relative to FEP+

However, I do have some concerns regarding the manuscript's writing and organization, which currently detract from the readability. Notably, there is no Table 3, yet there are two instances each of Table 4 and Table 5, which is quite confusing. I strongly encourage the authors to carefully review and correct these inconsistencies.

RESPONSE: We appreciate the referee for a positive evaluation and notice the issues with the tables, which we have revised and updated accordingly.